# Radio Channel Modelling for VHF System Operating in the Offshore Wind Farms Propagation Environment

**DOI:** 10.3390/s23177593

**Published:** 2023-09-01

**Authors:** Krzysztof Bronk, Patryk Koncicki, Adam Lipka, Rafał Niski, Błażej Wereszko

**Affiliations:** National Institute of Telecommunications, 04-894 Warsaw, Poland; k.bronk@il-pib.pl (K.B.); p.koncicki@il-pib.pl (P.K.); a.lipka@il-pib.pl (A.L.); r.niski@il-pib.pl (R.N.)

**Keywords:** wind farms, radio wave propagation, radio channel, marine radiocommunication, VHF band, signal interference

## Abstract

This article discusses several aspects related to the modeling of the radio channel in the wind farm propagation environment. The first part of the article is filled with the analysis of the ITU-R BT.1893-1 model, which—as will be shown—is also applicable to systems operating in the VHF and UHF bands. The conducted measurement campaign makes it possible to model a radio channel for VHF operating systems in the wind farm propagation environment. Thanks to the obtained results, the authors propose the VHF radio channel model for that environment. Using the software implementation of this model, the authors carry out a detailed simulation analysis of the impact of wind turbines on radio systems operating in the VHF band. The main motivation for this work is the growing importance of the offshore wind energy in general, numerous investments in that area in Poland, and the lack of existing mathematical models regarding the wind farms’ impact on the VHF systems. The VHF band is of special interest here, as this part of the spectrum is utilized by numerous marine administration and border protection radio systems, which can be adversely affected by the wind turbines (due to interference and/or radio shadowing).

## 1. Introduction

Renewable energy sources are developing much faster around the world than conventional energy sources, and among them the highest growth rate is wind energy. Currently, it provides approx. 7% of global electricity consumption, which makes it a world leader in green technologies. The latest statistics from the GWEC Global Wind Report for 2022 [1] show that the total power of all wind turbines in the world has now exceeded 837 GW. This corresponds to a growth rate of wind power production of 13% compared to 2020 (despite the overall economic disruption caused by the COVID-19 pandemic) and is a continuous upward trend compared to previous years. Such records in energy production by wind farms show a general development trend in the direction that ensures energy independence of countries and, thus, energy security. The current development of wind energy in the world takes place in two basic segments: onshore wind power and offshore wind power, where the latter direction has now become a sector with great development potential. The biggest advantages of the intensive development of offshore wind turbines include on the use of sea areas with higher wind speeds, resulting in higher efficiency; wind conditions at sea are more stable compared to onshore; the possibility of using turbines with higher power and, hence, sizes; and no controversy related to the installation of a power plant in the vicinity of inhabited areas. Turbines are being built further and further from the coast and they are getting bigger—both in terms of a single turbine, which has a higher mast, a larger blade span, a larger generator and, consequently, more power, and in terms of the size of wind farms with dozens or even hundreds of turbines. Offshore wind farms adopt larger and more clustered structures, which worsen the propagation conditions in such environment. This creates the need to analyze and evaluate the impact of the emerging offshore turbines on critical radiocommunication systems. In the case of offshore wind farms, the most important frequency range in the context of such analysis is definitely the so-called maritime VHF band (156–174 MHz), because it is utilized by most of the safety- and navigation-related systems (including AIS [2,3], GMDSS, VDES [4,5,6], and many other systems ensuring safety, security, and monitoring of shipping) operated by maritime administrations, navy, border guard, coastal guard, etc. (to a lesser extent, the medium frequency is also relevant to such analysis). For example, in Poland, any investor intending to build and operate an offshore wind farm on the Polish exclusive economic zone (EEZ) is obligated by law to submit an expert opinion discussing the possible impact of the planned farm on the critical maritime systems, and if any negative impact is anticipated, appropriate mitigation measures must be proposed. Such expert opinions must be submitted before any construction works even start and failure to do so means that the investor’s application will be rejected. This obviously only applies to specific Polish conditions, but at the same time it also amplifies the significance of this issue.

That being said, we have to make an observation that offshore wind farms’ impact on the VHF systems is not an easy matter to analyze. Very few sources discuss it let alone provide some theoretical/mathematical background for such a task. Obviously there exist numerous papers regarding channel modeling in the VHF band (see, e.g., [3,7,8,9,10]), but they usually concentrate on the typical applications of the VHF band, including maritime, military, or aeronautical communication, but hardly ever address the issue of VHF in the context of the wind farms. Consequently, their results cannot be easily “translated” (if at all) into the scenario that will be discussed in this article. On the other hand, a radar-based theory regarding the so-called radar cross section (RCS) of a wind turbine is known from many years of research and can provide some insight into the analysis of the wind farm impact on radiocommunication systems. In its base form, though, it applies to much higher frequency ranges than the maritime VHF Band. In 2011, the ITU-R issued the first version of the BT.1893 recommendation [11] which provides the mathematical background for the assessment of impairments caused by land wind farms, but it is originally dedicated to the digital television systems operating in the UHF Band (300–3000 MHz). This recommendation was expanded in the following years (see below). To sum up, no official standard exists which would serve as a background for the analysis of the offshore wind farms on the systems operating in the maritime VHF.

On the basis of the previous experience in the field, the authors of this paper formulated a hypothesis that possibly the above-mentioned ITU-R recommendation BT.1893 (theoretically dedicated to the UHF digital TV)—in connection with the RCS theory—could be “extended” to cover the maritime VHF systems as well. In order to verify this concept, we carried out dedicated measurement campaigns which will be discussed in the following paper. It should be noted that this “hypothesis” was based on substantial grounds rather than on a “blind guess”. In both the ITU-R recommendation and RCS theory, the key phenomenon is the signal reflection from the turbines (its blades, mast, etc.). Our understanding is that even though the wavelength in the maritime VHF is greater in comparison to UHF, and particularly SHF (radars)—which theoretically should reduce the impact of the reflection and other applicable phenomena such as absorption or scattering—this is not really the case. The offshore wind farms are generally much bigger constructions that land-based counterparts—that applies to all their components: masts, blade lengths, or blade widths. To put it in the simplest terms, offshore farms represent a much greater reflecting “surface”. Consequently, as far as the maritime VHF is concerned, not only do all the propagation effects relevant to land farms also occur on the offshore farms, they can be even more significant at sea. That was the main rationale behind our attempt to extend the theoretical models, originally created for land turbines and for higher frequencies, on the maritime VHF band. In the following sections, we will demonstrate our assumption was indeed right.

At the same time, it should also be noted that if the frequency in question gets very low this observation will not hold any more. For example, in the medium frequency band the wavelength becomes comparable with the physical size of the turbine, which definitely makes the reflecting effect practically negligible in this case.

The second part of the paper will be dedicated to yet another issue relevant to the offshore wind farm analysis. In order to more effectively plan and design critical VHF radio systems in the wind farm environment, there is a need to model a radio channel that will ensure the mapping of the impact of physical phenomena occurring in the real propagation environment. The authors of the article, on the basis of the measurement campaign results, proposed the parameters of the maritime radio channel model for wind turbines propagation environment, which aimed to ‘faithfully reproduce’ the impact of the phenomena relevant to the offshore wind farm areas. 

Moreover, the article describes the methodology adopted from the recommendation ITU-R BT.1893-1, which has been compared with the results obtained during the measurements. The realized measurement campaign has been discussed and the obtained results were presented in the context of modeling the radio channel for marine VHF systems operating in an offshore wind farm propagation environment. Finally, simulation analyzes for ship systems operating in the VHF band, performed using the newly created maritime wind turbine propagation channel model, were presented and discussed in the last part of the article.

To finish this introductory section and before going to the main body of this paper, it is worthwhile to once again list the key motivation factors for the research presented in the paper. These are as follows:

The growing importance of the offshore wind farms as a source of energy;A significant number of ongoing investments in Poland (the country of the paper’s authors);A lack of mathematical tools for modelling radio channel in the VHF (and UHF) bands in the wind farm environment;The significance of the VHF band with respect to the maritime applications. This is the band within which some of the most important systems utilized by marine administration (surveillance, safety, GMDSS) and/or border protection units operate. The proper operation of these systems can realistically be affected by wind turbines, mostly due to interference and radio shadowing;The wind turbines at sea are generally larger in size than their land counterparts. In the recent years, they have been “growing” even more (their size is becoming comparable with the wavelength), which translates to even stronger potential disruptions of the affected radio systems;The offshore wind farms’ investors in Poland are now obligated by law to present analysis of their farms’ impact on the selected maritime radio systems. It can be a problem given the previously mentioned lack of appropriate standards and guidelines.

To accomplish all the goals of this research, several obstacles had to be dealt with. The main challenges for the authors were as follows:The necessity of conducting highly complicated measurement campaigns (particularly challenging was maintaining synchronization in time and frequency);Potential impact of the land environment during the measurements;Lack of offshore wind farms on the Polish waters at the time of this writing (so measurements had to be conducted at land wind farms);No reference models, standards, etc., addressing the VHF channel in the wind farm environment.

In the authors’ opinion the major original contributions and output of this work include to following:The measurement verification of the ITU-R BT.1893-1 recommendation;The original channel model for the VHF band in the wind farm environment;Lessons learned on the effective planning and deployment of maritime radio systems in such environments, while minimizing the potential interference and radio shadowing effects and maximizing the ranges of these systems;The simulation tools and measurement equipment allowing multi-faceted analyses of the offshore wind farms impact on radio system, both in Poland and in other countries;Identification of the problem (with the remedy proposal), which will increase as the number of existing offshore wind farms grows and the size of the turbines grows.

## 2. Methods

### 2.1. ITU-R BT.1893-1 Model

For the purposes of researching wind turbines’ impact on radio communication systems and the possibility of correct modeling of the radio channel, two methods were adopted, described in recommendation ITU-R BT.1893-1 “Assessment methods of impairment caused to digital TV reception by wind turbines” [11].

The first of these methods—described in Annex 1 of the document—models the propagation for a single wind turbine, but when applied separately to individual turbines it also allows us to analyze the entire wind farms (as the entire wind farm effectively constitutes a “network” of interference sources, the methods for the cumulative effects calculation, such as the k-LMN, can well be utilized here). As will be shown in the subsequent parts of this paper, including the section describing the measurement campaigns, the Annex 1 model particularly allows us to address the reflection from the turbines’ rotors, which makes its very useful for far-field conditions. If the distance between the turbine and the signal source is significant, it is the reflection from the rotors that is by far the most dominant effect. On the other hand, the model included in Annex 2 of the recommendations [11], while also applicable both for individual turbines and for entire wind farms, is much more suitable for the near-field condition. The reason for that is that, unlike the Annex 1 model, in Annex 2 the reflection from the turbine’s tower (mast) is the major reflection scenario. If the distance between the turbine and the signal source is satisfying the near-field criterion, a much greater “portion” of this signal will reflect from the tower, rather than from the rotor (due to very shallow transmission angle with respect to the rotor). As a result, the Annex 2 method will be applicable to the studies of the impact of the farm on ship terminals, which may be located in close proximity to the turbines, while for the scenarios where the distance between the transmitter and the turbines is significant, the method from Annex 1 should be used (Please note that, theoretically, Annex 2 can also be applied for the far field scenarios, but for the reasons explained above, its results will be less reliable compared to those derived from the Annex 1). Those applicability guidelines regarding the two Annexes from the ITU-R 1893-1 recommendation—proposed by the authors—is one of the important outputs of this paper and it will be discussed more thoroughly in the subsequent sections.

#### 2.1.1. Wind Turbine Impact Studies on Radiocommunication Systems—Model ITU-R BT.1893-1 Annex 1

This recommendation generally concerns the impact of turbines on DVB-T digital television systems; however, as was proven through a measurement campaign organized back in 2015, this model is also applicable for radio communication systems operating in the VHF band. This model is facilitated by a mathematical description of the primary propagation mechanism at wind farm locations, i.e., the reflection of the radio signal from the wind turbine blades. The most important elements of the Annex 1 model are presented below. Please note that the verification measurement campaign results can be found in article [12].

Let us now assume the arrangement of the transmitter, receiver, and wind turbine as depicted in Figure 1.

Let us assume the distance between the receiver and the wind turbine is *r* (see Figure 1), the scattering coefficient, *ρ*, which includes the free-space loss for the path between the turbine; and the receiver can be expressed as follows [11]:(1)$$\rho =\frac{A}{\lambda r}g(\theta )$$
where
(2)$$g\left(\theta \right)=sinc^{2}\left[\frac{S}{\lambda }\left(cos\theta -cos\theta_{0}\right)\right]sin\theta$$
and
$$sinc\left(x\right)=\frac{\sin\left(\pi \cdot x\right)}{\pi \cdot x}.$$

*A*—total area of the turbine blades [m^2^];*S*—mean width of the blade [m];*λ*—radio signal wavelength [m];*r*—the distance between the wind turbine and the receiver [m];*θ*—the angle between the receive direction and the plane of the rotor, i.e., the angle of the signal reflected (scattered) from the blades [°];*θ_0_*—the angle between the transmit direction and the plane of the rotor, i.e., the angle of the incident signal at the blades [°].

The *g(θ)* function has values in the range of [–1, 1].

It might be stated that the coefficient *ρ* (1) represents the amount of the incident signal that will be reflected from the blades towards the receiver. 

The value of the *ρ* coefficient is at its maximum when the transmitter, receiver, and wind turbine are all in the same line and when, additionally, this line is perpendicular (normal) to the rotor’s plane. In such a case [11]
(3)$$\rho =\rho_{max}=\frac{A}{\lambda r}.$$

The FSWT (Field Strength at the Wind Turbine) parameter can be defined as the strength of the signal directly at the wind turbine location [11]:(4)$$FSWT=EIRP-L_{l}\;.$$

*EIRP*—equivalent isotropical radiated power of the transmitter [dBm];*L_l_*—propagation loss (attenuation) on the path between the transmitter and wind turbine (length l) [dB].

If the length of the path between the receiver and wind turbine is *r*, then the unwanted signal power (i.e., the power of the signal that propagates from the transmitter to the receiver due to reflection from the turbine blades) can be calculated as [11]
(5)$$UFSR=FSWT+20\log\left(\rho \right).\;$$

The UFSR (Unwanted Field StRength) is the parameter which helps analyze the wind farm as the source of a secondary radiation.

#### 2.1.2. Wind Turbine Impact Studies on Radiocommunication Systems—Model ITU-R BT.1893-1 Annex 2

The model presented in the Annex 2 of the recommendation [11] describes the radio channel in the presence of a wind farm in the form of the so-called Tapped-Delay Line, in which individual paths are characterized by the following:Corresponding delays;Mean amplitudes;Doppler spectrum associated to each path to account for the variability caused by the rotation of the blades.

The above parameters closely depend, among others, on the specific location of the transmitter and receiver with respect to the specific wind turbine. The key parameters of the Annex 2 model are presented further in this section.

Number of model paths

As a first approximation, the total number of model paths equals the number of turbines on the wind farm plus the direct path representing the signal from the transmitter. Depending on the values of delays and amplitudes determined in the further modeling procedure, this number can be ultimately reduced.

Relative delays of the path

For a given wind turbine, the relative delay *τ_i_(s)* of the scattered signal is determined as a function of the distance between the direct path (transceiver) and the scattered signal path (transmitter-> turbine-> receiver) [11]:(6)$$\tau_{i}=\frac{\left(R_{Tx-WT{}_{i}}+R_{WT_{i}-Rx}-R_{Tx-Rx}\right)}{c}.$$

*R_Tx-WTi_*—transmit antenna to i-th wind turbine distance [m];*R_WTi-Rx_*—i-th wind turbine to receive antenna distance [m];*R_Tx-Rx_*—transmit antenna to receive antenna distance [m];*c*—speed of light ≈ 300,000,000 m/s.

Mean amplitude of the paths

The direct path power from the transmitting antenna at the receiver location, *P_Tx-Rx_*, can be determined from the following relationship [11]:(7)$$P{}_{Tx-Rx}=\frac{P_{t}G_{Tx-Rx}G_{Rx-Tx}\lambda^{2}L_{prop}}{{\left(4\pi \right)}^{2}R_{Tx-Rx}^{2}}.$$

*P_t_*—maximum transmitter power [W];*G_Tx-Rx_*—transmit antenna gain in the direction of the receive antenna (dimensionless);*G_Rx-Tx_*—maximum gain of the receive antenna (dimensionless);*L_prop_*—propagation losses (dimensionless);*R_Tx-Rx_*—transmitter to receiver distance [m];*λ*—wavelength [m].

It is worth noting that the power described by the Formula (7) corresponds to the Annex 1 parameter, FSR (Wanted Field Strength). In turn, the power at the receiver of the scattered (reflected) signal from a given turbine, *P_Tx-WTi-Rx_*, can be calculated in the following way [11]:(8)$$P_{Tx-WT_{i}-Rx}=\frac{P_{t}G_{Tx-WT_{i}}G_{Rx-WT_{i}}\lambda^{2}\sigma_{i}}{{\left(4\pi \right)}^{3}R_{Tx-WT_{i}}^{2}R_{WT_{i}-Rx}^{2}}.$$

*G_Tx-WTi_*—transmit antenna gain in the direction toward the wind turbine *i* (dimensionless);*G_Rx-WTi_*—receive antenna gain in the direction toward the wind turbine *i* (dimensionless);*σ_i_*—bistatic radar cross section (RCS) of the mast in the receive antenna direction [m^2^];*R_Tx-WTi_*—transmit antenna to *i*-th wind turbine distance [m];*R_WTi-Rx_*—*i*-th wind turbine to receive antenna distance [m].

It is noteworthy that the power of the spread signal described by formula (8) corresponds to the Annex 1 parameter, UFSR (Unwanted Field Strength).

Doppler Spectra

In order to take into account the characteristics of the Doppler spectra, the authors of the ITU-R model [11] defined three representative, empirical cases of the Power Spectral Density (PSD), valid for the scenarios of high-, medium-, and low-degree time variability, which correspond to the rotational speed of the rotors and their spatial orientation. The maximum Doppler frequency, *f_Bmax_* (Hz), is defined by the recommendation as follows [11]:(9)$$f_{Bmax}=\frac{2\omega_{max}l}{\lambda }cos(\frac{\phi_{r}}{2}).$$

*ω_max_*—the maximum rotation frequency of the wind turbine [rad/s];*l*—blade length [m].

In the case of wind turbines, the maximum frequency *f_Bmax_* corresponds to the situation where *ϕ_r_* = 0° and the turbine blades rotate in the transmitting antenna-turbine-receiving antenna plane. In order to actually take into account the different wind conditions (which are likely to apply to different receiver locations), it is recommended that the three values of the Power Spectral Density (*S_high_*, *S_medium_*, *S_low_*) be determined below. The case of high variability of conditions (*S_high_(f)*) represents the most pessimistic case.
(10a)$$S_{high}\left(f\right)=\left\{\begin{array}{c} 19.7\exp(4.5\cdot \frac{f}{f_{Bmax}})-38.0\;\;\;\;\;\;-0.9\cdot f_{Bmax}\leq f<0\\ \delta \left(f\right)\;\;\;\;\;\;\;\;\;\;\;\;\;\;\;\;\;\;\;\;\;\;\;\;\;\;\;\;\;\;\;\;\;\;\;\;\;\;\;\;\;f=0\\ 21.4\exp\left(-4.8\cdot \frac{f}{f_{Bmax}}\right)-38.1\;\;\;\;\;\;\;0<f\leq 0.9\cdot f_{Bmax} \end{array}\right.,$$
(10b)$$S_{medium}\left(f\right)=\left\{\begin{array}{c} 22.0\exp(6.1\cdot \frac{f}{f_{Bmax}})-30.4\;\;\;\;\;\;-0.7\cdot f_{Bmax}\leq f<0\\ \delta \left(f\right)\;\;\;\;\;\;\;\;\;\;\;\;\;\;\;\;\;\;\;\;\;\;\;\;\;\;\;\;\;\;\;\;\;\;\;\;\;\;\;\;\;f=0\\ 25.1\exp\left(-8.7\cdot \frac{f}{f_{Bmax}}\right)-29.5\;\;\;\;\;\;\;0<f\leq 0.6\cdot f_{Bmax} \end{array}\right.,$$
(10c)$$S_{low}\left(f\right)=\left\{\begin{array}{c} 22.9\exp(17.9\cdot \frac{f}{f_{Bmax}})-24.9\;\;\;\;\;\;-0.3\cdot f_{Bmax}\leq f<0\\ \delta \left(f\right)\;\;\;\;\;\;\;\;\;\;\;\;\;\;\;\;\;\;\;\;\;\;\;\;\;\;\;\;\;\;\;\;\;\;\;\;\;\;\;\;\;f=0\\ 23.2\exp\left(-17.6\cdot \frac{f}{f_{Bmax}}\right)-25.0\;\;\;\;\;\;\;0<f\leq 0.3\cdot f_{Bmax} \end{array}\right..$$

*f_Bmax_*—defined by the Formula (9);*δ(f)*—is the Dirac delta function, defined as follows:(11)$$\delta \left(f\right)=\left\{\begin{array}{c} 0,\;\;\;f\neq 0\\ +\infty ,\;\;\;f=0 \end{array}\;.\;\;\;\;\;\;\;\right.$$

#### 2.1.3. Effective Reflecting Surface

The effective reflecting surface is known as the Radar Cross Section (RCS) as this concept is derived from radar technology. The RCS parameter describes the ability of an object to reflect radio (radar) or sound waves. It is expressed as the area (in [m^2^]) situated in a plane perpendicular to the direction of the incident wave, which, in the case of ideal, isotropic reflection of all the energy incident on it, would produce the same equivalent isotopically radiated power as the power generated by the real object. Factors that influence the RCS value include the following [13]:The size of the cross-sectional area of the object;The shape of the object;Material the object is made of or covered with;Frequency of the signal reflected from the object;Angle of incidence of the signal;Signal reflection angle;Polarization of the transmitted and received signal in relation to the target.

Objects with a larger effective reflection surface will reflect waves better, i.e., give a stronger echo. They can, therefore, be detected from a greater distance.

In electromagnetic analysis, the formula for the RCS parameter (most often denoted by the σ) can be presented as follows [14]:(12)$$\sigma =\underset{r\to \infty }{\lim}4\pi r^{2}\frac{{\left|E_{s}\right|}^{2}}{{\left|E_{i}\right|}^{2}}.$$

$E_{i}$—the intensity of the electric field incident the object;$E_{s}$—the intensity of the electric field scattered on the object.

Depending on the distance of a transmitter from the wind turbine, an appropriate mathematical formula is used to calculate the RCS parameter value. If the far-field criterion is met, the mathematical formulas for Annex 1 are used to calculate the RCS parameter [14,15,16] (since the Annex 1 is more suitable for large distances between the transmitter and turbine). In this case, the reflection of the signal from the wind turbine rotors is taken into account, where the calculation should primarily take into account the equivalent area of the blades of the wind turbine. However, if the far-field criterion is not met, the formulas in Annex 2 are used for calculations, where mainly reflection from the wind turbine mast, characterized by its radius, is taken into account.

In general, the effective reflecting surface for Annex 1 can be described by the following formula:(13)$${RCS}_{A1}\left[dBm^{2}\right]=10\log\left(\frac{4\pi A^{2}}{\lambda^{2}}\right)+10\log\left(f_{v}\left(\theta \right)\right)+10\log\left(f_{h}\left(\theta \right)\right)+L_{mat}\left(\varepsilon_{r}\right)$$
where lower index _A1_ indicates the formula is true for Annex 1 of [11].

*A*—equivalent of the total area of the blades [m^2^];*λ*—wavelength [m];*S*—mean width of the wind blade [m];$f_{h}\left(\theta \right)$—rotor’s equivalent antenna pattern (The wind turbine is treated as a “radio transmitter” generating interference, while the functions $f_{v}\left(\theta \right)$ and $f_{h}\left(\theta \right)$ indicate the directional characteristics of the “transmitting antenna”) in the horizontal plane;$f_{v}\left(\theta \right)$—rotor’s equivalent antenna pattern in the vertical plane;$f\left(\theta \right)={g(\theta )}^{2}$;$g(\theta )=sinc^{2}\left[\frac{S}{\lambda }\left(\cos\theta -\cos\theta_{0}\right)\right]\sin\theta$, where: $sinc\left(x\right)=\frac{sin(\pi x)}{\left(\pi x\right)}$;$L_{mat}(\varepsilon_{r})$—losses related to the relative permeability of the turbine blade material.

The graphic interpretation of the angles referred to in Formula (13) is presented in Figure 2 [11].

To compare the RCS parameters for both annexes, the losses resulting from the material and the antenna angles were temporarily omitted. Ultimately, the following formula was reached for Annex 1:(14)$${{RCS}_{A1,MAX}[dBm^{2}]=\sigma }_{A1,\;\;MAX}\left[dBm^{2}\right]=10\log\left(\frac{4\pi A^{2}}{\lambda^{2}}\right)+L_{mat}\left(\varepsilon_{r}\right).$$

To discuss the influence of the material the blades of the wind turbine are made from on the signal reflection, the $L_{mat}$ coefficient will be introduced. For a broader analysis of this issue, the concept of the wave reflection coefficient should be presented, i.e., the ratio of the intensity of the reflected wave to the intensity of the incident wave. An electromagnetic wave falling on the boundary surface of two media with given refractive indices is refracted and partially reflected. In general terms, the wave reflection coefficient can be described by the following formula [17]:(15)$$L_{mat}[dB]=20\log\left(|\gamma |\right),$$
where
(16)$$\gamma =\frac{Z-Z_{0}}{Z+Z_{0}}.$$

*Z* is the wave impedance of the electromagnetic wave, which in the case of an ideal dielectric (where the conductivity is close to 0) can be described as
(17)$$Z=\sqrt{\frac{\mu }{\varepsilon }}$$
where *μ* is the magnetic permeability and *ε* is the electrical permittivity.

$Z_{0}$ in the Formula (16) denotes the wave impedance in free space and is expressed by
(18)$$Z_{0}=\sqrt{\frac{\mu_{0}}{\varepsilon_{0}}}\approx 120\pi \Omega \approx 376.7\Omega .$$

Using the above formulas, it is possible to determine the value of the reflection wave coefficient applicable to the analyzed case. Modern wind turbine blades are usually made of glass fibers or glass-fiber-reinforced plastics (GFRP) (e.g., [18]). In order to calculate the value of the reflection wave coefficient, information was found that the relative permittivity of the GFRP material takes values $\varepsilon_{r}=3.3÷4.2\;$ [19]. In an isotropic, homogeneous dielectric with negligible magnetic properties, $\mu =\mu_{0}$ = 4π × $10^{-7}$ H/m and $\varepsilon =\varepsilon_{r}\times 8.854\times 10^{-12}\;F/m$. This means that the value of the wave impedance (in a isotropic and homogeneous dielectric with negligible magnetic properties) can be calculated from a simplified formula:(19)$$Z=\sqrt{\frac{\mu }{\varepsilon }}=\sqrt{\frac{\mu_{0}}{\varepsilon_{0}\varepsilon_{r}}=}\frac{Z_{0}}{\sqrt{\varepsilon_{r}}}=\frac{377}{\sqrt{\varepsilon_{r}}}\Omega ,$$

For GFRP we can assume $\mu =\mu_{0}$ [20], and the range of the reflection wave coefficient is
$${L_{mat}}_{\varepsilon_{r}=3.3}\left[dB\right]=20\log\left(\left|\gamma \right|\right)=20\log\left|\frac{Z-Z_{0}}{Z+Z_{0}}\right|=20\log\left|\frac{\frac{377}{\sqrt{3.3}}-Z_{0}}{\frac{377}{\sqrt{3.3}}+Z_{0}}\right|=-10.97,\;{L_{mat}}_{\varepsilon_{r}=4.2}\left[dB\right]=20\log\left(\left|\gamma \right|\right)=20\log\left|\frac{Z-Z_{0}}{Z+Z_{0}}\right|=20\log\left|\frac{\frac{377}{\sqrt{4.2}}-Z_{0}}{\frac{377}{\sqrt{4.2}}+Z_{0}}\right|=-9.25.$$

Modern turbine blades are often made of GFRP material with *ε_r_* = 4.2, for which the reflectance is approx. 9 dB. In the performed calculations, it was assumed that the theoretical reflection would be reduced by 9 dB.

The parameter $\sigma_{A2}$ (RCS of the turbine mast towards the receiving antenna) for Annex 2 can be calculated from the following equation [11]:(20)$$\sigma_{A2}\left(\phi_{r},\;\theta_{t}\right)=krL_{nf}^{2}\sqrt{\left(\frac{1+cos\phi_{r}}{2}\right)}\cdot sin\theta_{t}$$
lower index _A2_ indicates the formula is true for Annex 2 of [11],

$k=\frac{2\pi }{\lambda }$ (1/m).*r*—tower radius (m);*ϕ_r_*—receive antenna angular position in the horizontal plane measured at the wind turbine under consideration in an anti-clockwise direction from the direction of the transmit antenna;*θ_t_*—transmit antenna angular position in the vertical plane.The graphic interpretation of the angles referred to in Formula (20) is presented in Figure 3.

It should be noted that for distances at which the effect of the turbines may be significant, the far field criterion is usually not met, as in general [11]
(21)$$R_{Tx-WT_{i}}<\frac{2L^{2}}{\lambda }$$
where *L* gives the tower length (m).

In this case, the near-field scattering effects should be taken into account when considering the near-field tower height, *L_nf_* [m] [11]:(22)$$L_{nf}=\sqrt{\frac{\lambda R_{Tx-WT_{i}}}{2}}.$$

The inequality specified in Equation (21) may effectively be considered as the upper bound for the application (Please note this is merely a recommendation taking into account the specifics of the two Annexes and the conclusions from the measurements conducted by the authors of this paper. Formally. the Annex 2 model can be applied for far-field as well; however, it will be much less reliable in such a case compared to Annex 1) of Annex 2. If this inequality is no longer true (for higher values of *R_Tx-WTi_*), Annex 1 should be applied instead.

The mean amplitude of each path can be determined as the ratio of the power of the reflected (unwanted) component to the direct (wanted) component [11]:(23)$$P_{i}=10\log\left(\frac{P_{Tx-WT_{i}-Rx}}{P_{Tx-Rx}}\right).$$

Components for which the mean *P_i_* amplitude is less than −45 dB should be ignored (as insignificant). 

To compare the RCS formula for Annex 2 with the formula for Annex 1 (13), we can convert the Equation (20) to
(24)$${RCS}_{A2}\left[dBm^{2}\right]=\sigma_{A2,\;\;MAX}\left[dBm^{2}\right]+10\log\left(f_{A2}\left(\phi_{r}\right)\right)$$
where
$$f_{A2}\left(\phi_{r}\right)=\sqrt{\left(\frac{1+cos\phi_{r}}{2}\right)}\cdot sin\theta_{t}$$
and
(25)$${{RCS}_{A2,MAX}[dBm^{2}]=\sigma }_{A2,\;\;MAX}\left[dBm^{2}\right]=10log(krL_{nf}^{2}).$$

Figure 4 shows the relationship between the RCS parameter (for both Annexes of [11]) and the distance between the wind turbine (137 m height) and the transmitter. For turbine 1, the equivalent area of wind turbine blades *A* = 400 $m^{2}$ and the mast radius *r* = 3.2 m were assumed. For turbine 2, *A* = 700 $m^{2}$, and *r* = 4 m were assumed.

For Annex 1, RCS takes a constant value because it does not depend on distance. On the other hand, for Annex 2, substituting the relation (22) to the Formula (25), one can notice a change in the RCS value [21] depending on the distance between the turbine and the transmitter. Based on the presented analyzes, the Formula (8) can be clarified and for Annex 1 we obtain
(26)$$P_{{UFSR}_{{Tx-WT_{i}-Rx}_{RCS1}}}=\frac{P_{t}G_{Tx-WT_{i}}G_{Rx-WT_{i}}\lambda^{2}\sigma_{A1,MAX}}{{\left(4\pi \right)}^{3}R_{Tx-WT_{i}}^{2}R_{WT_{i}-Rx}^{2}}=\frac{P_{t}G_{Tx-WT_{i}}G_{Rx-WT_{i}}A^{2}}{{\left(4\pi \right)}^{2}R_{Tx-WT_{i}}^{2}R_{WT_{i}-Rx}^{2}}$$
and for Annex 2
(27)$$P_{{{UFSR}_{{Tx-WT_{i}-Rx}_{RCS2}}}_{}}=\frac{P_{t}G_{Tx-WT_{i}}G_{Rx-WT_{i}}\lambda^{2}\sigma_{A2,MAX}}{{\left(4\pi \right)}^{3}R_{Tx-WT_{i}}^{2}R_{WT_{i}-Rx}^{2}}=\frac{P_{t}G_{Tx-WT_{i}}G_{Rx-WT_{i}}\lambda^{2}r}{{4\left(4\pi \right)}^{2}R_{Tx-WT_{i}}R_{WT_{i}-Rx}^{2}}.$$

Thanks to this transformation, the formulas for the maximum UFSR value for Annexes 1 and 2 were obtained.

### 2.2. Modelling of the Radio Channel Based on Selected Parameters

Signal propagation in a radio channel is influenced by many factors. It depends primarily on the properties of the wave itself, i.e., its length and polarization, as well as on the conditions of the environment in which the wave propagates. By “environmental conditions” we mean topography and the type of coverage—radio waves propagate differently in areas near water, in forests, in urban or open areas. In the area covered by this article, communications with sea-going vessels are mainly based on the VHF band systems. Ultra-high frequency waves propagate in the form of a ground-level spatial wave. In practice, at most a quasi-horizontal range of about 30 national miles (Nm) can be achieved. It should be emphasized that in the maritime radio channel the terrain does not affect wave propagation; however, reflections from the sea surface must be taken into account, as well as the possibility of the first Fresnel zone being obscured by the water surface, which is also affected by the curvature of the Earth. This causes multipath propagation, as a result of which one or more reflected waves reach the receiver. In turn, multipath is a source of interference and transmission quality degradation. The propagation of the signal in the VHF marine radio channel is also influenced by tropospheric refraction, which may increase the range due to the formation of tropospheric ducts [22]. Multipath propagation causes the signal—through diffraction, refraction, scattering, and reflection—to reach the receiver as a sum of component signals with different delays, phases, and amplitudes. The main parameters that describe a multipath propagation radio channel are as follows:Power Delay Profile (PDP)—determines how the strength of the signal reaching the receiver input changes over time. In general, it is the impulse response of the channel to an excitation with a very short pulse, defined as a series of components with different power levels and different delays (Figure 5); the delay of the *k*-th component is denoted as $\tau_{k}$.

To describe such a profile, we usually use the following parameters [23]:(28)$$\mathrm{average}\;\mathrm{delay}:\;\bar{\tau }=\frac{\sum_{k}P(\tau_{k})\tau_{k}}{\sum_{k}P(\tau_{k})},$$
(29)$$\mathrm{standard}\;\mathrm{deviation}:\;\sigma_{k}=\sqrt{\bar{\tau^{2}}-{\left(\bar{\tau }\right)}^{2}}.$$

Coherence bandwidth (*B_c_*)—means the frequency range in which the channel can be considered invariant. In this range, the channel amplitude response is constant and the phase response is linear. This means that the channel will similarly affect two sinusoidal signals separated by less than *B_c_* [Hz] (that are inside the coherence bandwidth), while two signals spaced apart by a bandwidth greater than *B_c_* will be affected differently. The *B_c_* band is often defined as the frequency range where the frequency correlation is greater than 0.5. In this case [23,24],
(30)$$B_{c}\approx \frac{1}{5\sigma_{\tau }}$$
where $\sigma_{\tau }$ it is the maximum spread of the delay.Doppler Spread (or Doppler band—*B_D_*)—determines the extent to which the band has expanded as a result of one of the radio system components (transmitter or receiver) being in motion. As it is known, during the movement of the mobile station relative to the base station, the frequency of the received signal will differ from the frequency of the transmitted signal. The frequency deviation depends on the speed and the angle of the electromagnetic wave. The Doppler band is defined as the maximum Doppler shift, i.e.,
(31)$$f_{d\;max}=\frac{v}{\lambda }$$
where *v* is the speed of the terminal and λ represents the electromagnetic wavelength. If the baseband of the signal is much greater than *B_D_*, the Doppler effect can be ignored.
Coherence time (*T_c_*)—defines the period of time during which the channel remains invariant as a function of time (i.e., the time during which the impulse response of the channel is approximately constant). Channel coherence time is approximately the inverse of the Doppler band [23]:
(32)$$T_{c}\approx \frac{0.423}{B_{D}}=\frac{0.423}{f_{d\;max}}.$$

Based on the above parameters, four basic types of radio channels can be defined:Flat fading channel—a frequency non-selective channel, the characteristics of which are constant throughout the entire band. Fading that can affect the entire band is called flat fading. This type of channel occurs when
(33)$$B_{s}\ll B_{c}\;\;\;\;\;\;\;or\;\;\;\;\;\;\;T_{s}\gg \sigma_{\tau }$$
where *B_s_*—signal bandwidth and *T_s_*—symbol duration.
Frequency selective fading channel—a channel that influences some frequency components more strongly than others. This happens when
(34)$$B_{s}>B_{c}\;\;\;\;\;\;\;or\;\;\;\;\;\;\;T_{s}<\sigma_{\tau }.$$
This type of channel introduces intersymbol interference (ISI).
Fast fading channel—the channel whose impulse response changes faster than the duration of the symbol. This is due to the displacement of one of the system components (Doppler spread). For a channel with fast fading, the following relationships are true:
(35)$$T_{c}<T_{s}\;\;\;\;\;\;\;or\;\;\;\;\;\;\;f_{d\;max}>B_{s}.$$
Slow fading channel—a channel for which the relations are opposite to those for point 3; i.e., the Doppler spread is much smaller than the signal bandwidth:
(36)$$T_{c}\gg T_{s}\;\;\;\;\;\;\;or\;\;\;\;\;\;\;f_{d\;max}\ll B_{s}.$$


Out of the above four cases, the most unfavorable impact on the resultant transmission quality occurs during the situation when the channel is frequency-selective and, additionally, it is characterized by fast fading. Selectivity introduces the need to implement complex channel characteristic equalizers, usually in the form of a filter. On the other hand, when a channel has fast fading, information about its state must be transmitted at high frequency. When the channel changes faster than the frequency of the channel estimation, the power control or adaptive modulation algorithms may not be efficient enough. 

As part of this article, a measurement campaign was carried out in the vicinity of a selected onshore wind farm, which allowed us to determine the power delay profile of the channel power taking into account the reflections from individual wind turbines. The coherence time and the channel coherence bandwidth were also determined. In the further part, based on the developed software, the results of the simulation calculations of the basic parameters of the radio channel for a wind farm located in the marine environment are also presented. The obtained results can be verified in the future during the measurement campaign organized in the vicinity of the offshore wind farm.

### 2.3. The Measurement Campaign

For the purposes of the research conducted by authors, a measurement campaign was organized in October 2021. From the beginning it was clear that it would be implemented on an onshore farm (this is because, as of 2022, no offshore wind farm exist yet on the Polish waters). So, one of the criteria was to look for a farm where the turbines have the following characteristics: (a) Are as high as possible; (b) Have as large of a reflecting surface as possible. The idea was to provide a good approximation of the conditions prevailing in offshore wind farms (although it is clear that, as a rule, offshore wind farms use much larger turbines than the farms located on land). In the course of the analyzes, the Jasna wind farm located in the northeastern part of Poland, near the city of Malbork, was selected for the measurement campaign. In order to try to compare the levels of the measured signals with the mathematical models, it was necessary to obtain certain physical parameters of the wind turbines used in the Jasna wind farm. Table 1 summarizes that input data.

Two types of measurements were carried out as part of the research on radio channel modeling for systems operating in the VHF band in the propagation environment of wind farms:Measurements of the signal levels reflected from the rotors of wind turbines using the pulse signal;Doppler spectra measurements using signal carrier transmission.

#### 2.3.1. Measurements of the Signal Levels Reflected from the Rotors of Wind Turbines Using the Pulse Signal

The purpose of this part of the measurements was primarily to obtain a power delay profile, which is one of the parameters modeling the radio channel, and to verify the applicability of the ITU-R BT.1893-1 model in Annex 1 and Annex 2 versions. During the measurements, two professional, calibrated measurement sets were used: transmitting and receiving. The transmitting set consisted of the following (see Figure 6a):Signal generator (250 kHz–6 GHz): Agilent N5182A MXG Vector Signal Generator with the pulse modulation capability;Transmitting directional antenna: SIRIO WY155-3N (155–175 MHz);Power amplifier: Popek Elektronik type PEA02-2-50 with reflectometer;Two measurement cables: 3-m cable RG-214 N-N;Rubidium oscillator Quartzlock E80-GPS (+GPS antenna).The receiving set comprised the following parts (see Figure 6b):Spectrum analyzer: Keysight FieldFox N9914A with the appropriate options;Receiving antenna: Emco Model 3121C (28 MHz–1 GHz);Measurement cable: 3-m cable Huber+Suhner ST18A/11N468/11N468/3000MM;GNSS RTK Receiver GINTEC M1G2.

Measurements were made for the frequency of 161.8375 MHz. The pulse parameters and the mutual distances between the transmitter, receiver, and the wind turbine were selected in such a way that it was possible to distinguish the reflected signals from individual wind turbines using the presented above measuring equipment. To ensure full synchronization of transmitting and receiving, a reference PPS (Pulse Per Second) signal was used, generated on the transmitting side by the Quartzlock E80-GPS disciplined rubidium oscillator (whose starting point is synchronized with the beginning of the UTC second), while on the receiving side, the PPS signal was generated by the GNSS RTK Receiver GINTEC M1G2. The length of the generated impulse signal was set at 250 ns and it was repeated every second (according to the beginning of the UTC second). This value is due to the maximum hardware capabilities. The parameters of measurement antennas are as follows:Receiving antenna—dipole Emco 3121C set, calibrated for 161.8375 MHz band, 1.4 dBi gain;Directional transmitting antenna—SIRIO WY155-3N, 7 dBi gain on the main beam direction and 16 dB backward attenuation.

It is worth adding here that during all measurements carried out as part of the campaign, both the transmitting and receiving antennas were placed approximately two meters above ground level and the transmitting antenna was pointing at the nearest wind turbine. The transmitter–receiver line had to be parallel to the day’s wind direction in order to identify a rotor whose plane is currently perpendicular to the impulse direction (and this was the targeted turbine rotor).

#### 2.3.2. Doppler Spectra Measurements Using Signal Carrier Transmission

The measurements of the Doppler spectrum were performed using the same measuring station as for the measurements using impulse transmission, with one difference: the carrier signal was transmitted in a continuous mode. Therefore, there was no need to perform time synchronization of the transmitting and receiving station, while the carrier frequency was disciplined on both sides: in the signal generator (on the transmitting side—using the rubidium oscillator) and in the spectrum analyzer (on the receiving side—using GPS option). The measurement studies described in this section were aimed at modeling the radio channel to characterize the signal propagation in the presence of a wind farm. Having achieved that, the radio signal propagation channel in the presence of wind turbines can be modeled with the Doppler spectrum associated with each path to account for variability due to blade rotation.

Doppler spectrum measurements were used to determine the frequency and shape of the Doppler spectrum. The Doppler frequency depends on the relative positions of the transmitting antenna, wind turbine and receiving antenna *ϕ_r_* (shown in Figure 3), the maximum rotation frequency of the wind turbine *ω_max_* (rad/s), and the blade length *l*, as per Equation (9). In the case under consideration, $f_{B\;max}$ corresponds to the angle *ϕ_r_* = 0° and the blades rotating in the plane transmitting antenna—wind turbine—receiving antenna. Substituting the parameters of the Vestas 126-3.45 MW turbines, utilized in the measurement campaign site (see next section):Maximum rotation speed: 16 rpm;Blade length: 61.7 m.

We obtain a maximum Doppler frequency of 111.5 Hz. Doppler power spectral density, in accordance with ITU-R BT.1893-1 (Annex 2), can be determined depending on the maximum Doppler frequency and wind conditions (low, medium, and high variability) translated into different rotational speed and orientation of turbine rotors.

## 3. Results and Discussion

### 3.1. The Measurement Results

The measurements discussed in this article were carried out in 26–28 October 2021 at the Jasna wind farm located in the northeastern part of Poland, near the city of Malbork. The most important results obtained during the research that allowed us to characterize the radio channel in the presence of a wind farm are presented below.

#### 3.1.1. Measurements of the Level of Signals Reflected from the Rotors of Wind Turbines

During the measurement campaign, three measurements were carried out (on 26, 27, and 28 October 2021), the purpose of which was to determine the level of the signal reflected from the rotors of wind turbines at the receiving point—thanks to which it was possible to obtain the power delay profile needed to model the radio channel. The measurements concerned the frequency of 161.8375 MHz, which is related to the fact that it is a frequency in the range of the so-called the VHF marine band (156–174 MHz), to which this article applies. On 26 and 27 October, the measurements were carried out on the northern part of the Jasna wind farm (Figure 7a), whereas on 28 October research was conducted on the southern part (Figure 7b). The general methodology of measuring the level of the reflected signals realized with the use of the impulse signal was described in the previous section. At this point, it should only be remembered that its essence was the emission of a pulse with a duration of 250 ns, directed towards a specific turbine. Of course, the impulse reflected not only from the target turbine but from other turbines as well. The signals reflected from the farm were visualized on the spectrum analyzer screen (in time domain using zero span function) in the form of successive peaks with increasing delays in relation to the moment of receiving the direct signal. Knowing the exact locations of each wind turbine within the farm and the exact location of the transmitter, receiver, and the corresponding distances, we are able to calculate (using the formula for the speed of an electromagnetic wave) the exact delays after which the signal reflected from a specific turbine should reach the receiver. Consequently, by analyzing the delay values of successive peaks on the spectrum analyzer, we can determine with high probability from which specific turbine the given component of the received signal came from. With this knowledge, this scenario—for a specific turbine—was “re-enacted” in the simulation tool, in which the theoretical reflected signal level (UFSR) was determined for the ITU-R BT.1893-1 model in the Annex 1 and Annex 2 versions [11], and then these theoretical levels were compared with the reflected signal values that were actually measured. 

Figure 7 shows the location of the transmitter and receiver for each measurement scenario against the background of the turbines of the north and south Jasna farms. For the first scenario, the location of the transmitter and receiver was marked with Tx1 and Rx1, respectively. Analogous notation was used for the subsequent measurement scenarios. Wind turbines are marked with red dots. Arrows mark the direction in which the impulse was pointed at the turbine.

Table 2 shows all distances at which measurements were made for the given scenarios. It includes the distances between the transmitter and receiver and their distances between the wind turbine at which the signal was aimed. The maximum theoretical UFSR was calculated for each case. The reflectance from Annex 1 is approx. 10 dB higher than the Annex 2 reflectance. It should be noted, though, the blades are made of GFRP material, which has been shown to reduce the reflection by approx. 9 dB. All this considered, the maximum theoretical UFSR levels are very similar for both Annexes.

Table 3 shows the results of the time domain measurement, for which the maximum measured level of the UFSR reflected signal was obtained in comparison with the calculated theoretical power delay profile. In the case of the first presented scenario, it can be determined that the signal reflection could have occurred from at least six wind turbine rotors (peaks marked in the screenshot with numbers 1, 2, 3, 4, 6, and 7; the first peak marked No. 5 represents the direct signal, the level of which is denoted by the acronym FSR). In that scenario, the transmitter and the receiver were in line with the turbine No. 8. 

Generally, the first “peak” in the diagram on the left is a direct signal and, in order to determine the delay, it is necessary to calculate the difference between the reflected signal and the first pulse and then compare this value with the second diagram (the direct signal is at point 0). For example, the first reflected signal in scenario 1 is delayed by 7.05 − 5.8 = 1.25 µs, which is in line with the theory. As can be seen, for other scenarios, the measured delays are also consistent with theoretical values.

Table 4 compares the measured results with the theoretical values derived from the ITU-R BT.1893-1 (Annex 1 and Annex 2 versions). 

In the second measurement scenario, the turbine at which the impulse was aimed was the turbine numbered 11, while in the third scenario of measurements there was the turbine numbered 7. The transmitter and the receiver were in line with the turbines. The measurement results compared with the theoretical values are shown successively for these scenarios in Table 5 and Table 6.

When analyzing the results of the individual measurement scenarios (Table 4, Table 5 and Table 6), it is worth noting, first of all, that on each of them the measured value of the direct signal was lower than it would appear from the models. More precisely, the difference between the measured value and the theoretical value of the FSR parameter for each of the three scenarios were −42.24 dB, −36.1 dB, and −33.7 dB, respectively. The observation of the direct component (but not its power) on the spectrum analyzer screen served only to determine the time reference point against which the delays of individual reflected signals were measured and then assigned to specific turbines. Moreover, the terrain profile between the transmitter and receiver during the selection of the measurement scenarios was not optimized; on the transmitter -> receiver route, there often were obstacles, which, combined with the low height of the antenna, resulted in a strong propagation attenuation. In the course of the simulation, the simplest propagation model (free space loss) was used to determine the FSR parameter, so the calculated attenuation value is much lower than it would appear in real conditions. The above conclusion, combined with the lack of precise information on the transmitting antenna’s back radiation suppression, mean that the simulated values of the FSR parameter are roughly approximate, at best. Analyzing the results from Table 4, Table 5 and Table 6, we can see that the most pessimistic results (i.e., those where the theoretical level of the reflected signal was most overestimated in relation to the actually measured one) in the case of Annex 1 were obtained:

In Scenario 1—for reflection from turbines 8, 11, and 13 (the calculated bias level of the model is equal to −10.64 dB, −13.34 dB, and −12.44 dB, respectively);

In Scenario 2—for reflection from turbines 11, 13, and 12 (the calculated bias level of the model is equal to −14.91 dB, −13.59 dB, and −20.86 dB, respectively);

In Scenario 3—for reflection from turbines 7 and 6 (the calculated bias level of the model is equal to −17.47 dB and −17.28 dB, respectively).

It was also noted that wherever Annex 1 resulted in unacceptably high positive values for UFSR bias, in case of Annex 2 the respective values were negative (pessimistic model) and kept at a moderate level from approx. −5 dB to −20 dB. This confirms the applicability of the Annex 2 model in this case. Further in this section, the authors will explain in which situations to use a specific model (Annex 1 or Annex 2) based on the ITU-R BT.1893-1 recommendation [11].

Based on the measurements and theoretical calculations using the ITU-R BT.1893-1 model (presented in Table 3, Table 4, Table 5 and Table 6), the coherence bandwidths for individual cases was determined and listed in Table 7. It should be emphasized here that the coherence bandwidth calculation for the measured radio channels was performed in two ways (using two different FSR values). The first one based on the measured FSR value (power of the signal received directly from the transmitting station). However, the FSR measured in this way is subject to an error due to the following: (a) The propagation between the transmitter and the receiver took place close to the ground, so the diffraction had a greater effect than in the case of reflection from the wind turbine; (b) Terrain obstacles occurred in the first Fresnel zone; (c) The receiving antenna was positioned in the opposite direction to the main radiation beam of the transmitting antenna. Consequently, the calculations of the FSR were later adjusted, assuming the power level received from the known transmitter at its main antenna beam and considering solely free space propagation losses. The result of this “recalculation” can be seen in the last column of Table 7 and in the first row of Table 4, Table 5 and Table 6 (the “Measurement” field).

Scenario 1 and 2 refer to measurements made in the northern part of the Jasna wind farm consisting of 25 wind turbines with a height of 117 m. The results obtained for both scenarios are similar: the theoretical coherence bandwidth is approx. 600 kHz, while the value calculated on the basis of the measurements is approx. 100 kHz. However, as already mentioned above, the measurement of the transmitted signal component arriving via the direct path to the receiver was disrupted. For this reason, further calculations were made based on the theoretical FSR value and the obtained values were approx. 2500 kHz in this case. Thus, it can be concluded that the actual coherence bandwidth, calculated on the basis of the measurement data, is clearly wider than the theoretical value calculated on the basis of the ITU-R BT.1893-1 model, which means that the model is pessimistic in this respect. The same applies to the results obtained for scenario 3, which refers to the measurements made in the southern part of the Jasna wind farm consisting of 14 wind turbines with a height of 137 m. For this scenario, all values of the coherence bandwidth were higher than those in scenarios 1 and 2, but the relations between them confirm the previously formulated observation.

#### 3.1.2. Measurements of the Doppler Power Spectra Density

The measurements of the Doppler power spectra density were performed within three measurement scenarios, in three different locations—two in the north part of the Jasna wind farm (Figure 7a) and one in its south part (Figure 7b). Wind conditions varied from day to day, both in terms of strength and direction; Table 8 summarizes the rotational speeds of the turbine rotors and the corresponding Doppler frequencies. In addition, a channel coherence time (Tc) was determined and shown in the table where the channel is invariant as a function of time (i.e., the time where the channel impulse response is approximately constant). The channel coherence time is (approximately) inversely proportional to the Doppler bandwidth, so if the symbols are shifted by more than Tc with respect to one another their time correlation will be below 0.5 (see formula 32 and [23]).

According to the methodology of determining the Doppler spectrum presented in recommendation ITU-R BT.1893-1 [11], three variants of power spectral density are defined depending on the degree of variability in time, the rotational speed of the rotors, and their spatial orientation (called low, medium, and high variability). They differ from each other in the shape of the spectrum and the level of its concentration around the carrier frequency. According to the model:

“High variability”—covers the range from $-0.9\cdot f_{B\;max}$ to $0.9\cdot f_{B\;max}$;“Medium variability”—covers the range from $-0.7\cdot f_{B\;max}$ to $0.6\cdot f_{B\;max}$;“Low variability”—covers the range from $-0.3\cdot f_{B\;max}$ to $0.3\cdot f_{B\;max}$.

Table 9 presents these frequency limits for all three discussed variants and three sets of measurement scenarios.

Figure 8 shows the theoretical calculations of the Doppler spectrum (for high, medium, and low variability variants from recommendation [11]) and measured for scenario 1. Table 10, Table 11 and Table 12 contain numerical values of the Doppler power spectrum determined computationally and measured for each scenarios. The presented results have been normalized so that they can be directly compared. 

By analyzing the measurement results and comparing them with theoretical calculations, a similar shape of the Doppler spectrum can be observed. Moreover, in the range of approx. +/− 10 Hz, the plots with the measurement data coincide with the theoretical plots for low variability, which are the widest around the carrier frequency of all theoretical plots. In the measurement campaign carried out at the wind farm, it was not possible to separate individual paths representing each of the wind turbines when measuring the Doppler spectrum. As a result, the Doppler spectrum measurements were conducted for the entire wind farm. Based on the obtained results and observations, appropriate conclusions were drawn and included in the article. It should be noted that the central part of the spectrum of the received signal, around the carrier frequency, corresponded to the direct reflection from the turbine rotor used as the main source of the reflected signal (towards which the transmitting antenna was directed) and also to the “static” part of the reflected signal, which may come from, e.g., the nacelle or the tower. The rest of the measurement graphs (above 10 Hz/below −10 Hz) are similar to the theoretical graphs for high variability, the spectrum of which extends farthest from the carrier frequency and at the same time is the most suppressed in relation to it. This situation is most likely also due to the fact that the signals reflected from neighboring wind turbines are also received. It should be mentioned that for the scenario 1 and 2, different rotational speeds of the rotors of neighboring wind turbines were observed in relation to the nearest turbine, which is the main source of the reflected signal. On the basis of the collected and analyzed measurement data, it should be stated that recommendation ITU-R BT.1893-1 illustrates the shape of the Doppler power spectral density very well. For the existing measurement conditions, it is stated that for frequencies from about −10 to +10 Hz the low variability version of the Doppler spectrum is the best representation, while outside this range the best approximation is the high variability version. The second parameter to be considered when modeling the radio channel is the maximum Doppler frequency. The value of the maximum Doppler deviation determined during the measurements was below 50 Hz, while the theoretical value calculated for the maximum rotational speed of the turbine (16 rpm) and blade length of 61.7 m was 111.5 Hz. The determined values should be related to the frequency stability of devices operating in the considered frequency band. In the VHF band, two types of transmission should be distinguished:

Digital—the AIS system;Analog—Voice VHF transmission.

According to the AIS system specification [28], the frequency stability of transmitters and receivers should be at least 500 Hz. On the other hand, in accordance with the recommendation ITU-R M.489-2 [29], the required frequency stability is at least 5 ppm for coastal stations and 10 ppm for ship devices. For the considered frequency range, these two values translate into a deviation of 809.1875 Hz and 1618.375 Hz, respectively.

The values of the Doppler deviation determined by measurement and calculated theoretically are much lower than the required frequency stability of the systems under consideration (AIS devices ensure correct operation of the system with a frequency deviation of up to 500 Hz). It should also be noted that the channel coherence time determined during the measurements was from 4.778 to 7.984 ms, while the symbol duration in the AIS system is 0.104 ms. The relation (36) is fulfilled; therefore, in the considered channel, the impulse response does not change during the symbol duration, which means that only slow fading occurs in the channel.

### 3.2. Radio Channel Model for Systems Operating in the VHF Band in the Propagation Environment of Offshore Wind Farm with the Use of the ITU-R Recommendation Methodology BT.1893-1

On the basis of the measurement campaign and the obtained results, the authors of this article have defined specific situations in which the appropriate models described in the ITU-R BT.1893-1 recommendation should be used. These guidelines can be very useful in the practical application of the developed channel model in simulation studies for wind farms in the marine environment.

#### 3.2.1. Dependence on the Location of the Transmitter and Receiver in Relation to the Specific Wind Turbine

In the Annex 1 model, the primary reflection mechanism is the radio signal reflection from the wind turbine blades. As a result, this model is generally the most reliable for scenarios where the signal reflection from the rotor is as close as possible to the maximum reflection, i.e., when the transmitter is approximately perpendicular to the rotor. In the case of the Annex 2 model, the basic reflection mechanism from the turbine is the reflection from its mast. Of course, when the signal is transmitted towards the rotor directed at an angle of 90° to the transmitter, the reflecting surface (i.e., the surface of the rotor blades) will be much larger than the surface of the mast—especially in the case of offshore farms. Therefore, it should be stated that for scenarios where the signal reflection is close to the maximum reflection, the Annex 1 model will generally be more pessimistic than the Annex 2 model, and in such situations the Annex 1 should be used.

In the case of reflection from other turbines (with more shallow angles), the bias (difference between measured and theoretical value) of the Annex 1 model in many cases is not only greater but, above all, its values are positive (biases ranging from +4 dB to +92 dB), which means that the model becomes overly optimistic and the power of the reflected signal calculated with its use is lower than actually measured. Relating these results to the location of the transmitter, receiver, and turbines (Figure 7), we can see, however, that in this case the situation concerns the reflection of the signal from the turbines located at a large (and often very large) angular distance from the direction of the maximum reflection. In this case, the Annex 1 model becomes unreliable, as we deviate significantly from the scenario for which the model was originally defined. In practice, however, such large, positive values of errors, should not come as a surprise. If the signal is reflected from the turbine at a shallow angle (several dozen degrees away from the direction of the maximum reflection), then the reflection takes place from the side-inverted rotor, i.e., representing an incomparably smaller reflecting surface compared to the rotor perpendicular to the transmitter. The values resulting from the Annex 1 model may, therefore, give very low values of the UFSR reflection level. In addition, since the measured value is so significantly higher than the theoretical value, it allows us to conclude that some reflection mechanism—other than the reflection from the blades—is more prevalent in this case (e.g., reflection from the mast).

Consequently, for turbines that are far (on an angular scale) from the line of maximum reflection, the Annex 1 model should not be used: the Annex 2 model is the most appropriate in this case. In Annex 2 the dominant reflection mechanism from the turbine is the reflection from its mast. The main distinction is that for rotors the size of the reflecting surface depends on their temporal orientation, but in the case of the mast such relation obviously no longer exists. The second scenario in which the Annex 2 model should be used is when the transmitter is in the vicinity of the turbine (it is worth noting that in this case the angle of incidence of the signal on the rotor will very often be away from an angle of 90 degrees). According to the ITU-R BT.1893-1 model, this distance should be a maximum of 14.765 km (for a 117 m high turbine) and 20.245 km (for a 137 m high turbine).

In addition, at the end of these analyzes, the mean value of the UFSR error for the Annex 1 and Annex 2 models was calculated; however, while doing so, we only considered those values that are consistent with the applicability of the specific model, i.e., in the case of Annex 1 only the values that result from reflections from turbines approximately in line with transmitter and receiver were included, and for the Annex 2 calculation the remaining values were taken into account. The final results are presented in Table 13.

For Annex 1, the theoretical level of UFSR resulting from the model was on average 15 dB higher than it was obtained from the measurements. In turn, the model from Annex 2 overstated the level of the reflected signal power by an average of 9 dB in relation to the measurement data. These values should be treated as a correction factor in the application of the ITU-R BT.1893-1 recommendation in measuring the impact of wind farms on systems operating in the VHF band. This means that both models are sufficient to assess the impact of wind turbines on communication systems operating in the 160 MHz band. It is worth mentioning here that in 2013 [12], while researching the impact of wind turbines on radio/radar systems, a similar analysis was performed on the basis of the measurement data obtained at that time. In 2013, recommendation ITU-R BT.1893-0 contained only the Annex 1 model, but the results obtained at that time—with regard to Annex 1 only—are in line with the present ones.

#### 3.2.2. Dependence on the Frequency Range in the Model of Determining the Doppler Power Spectral Density

The next criterion for modeling the radio channel for VHF systems was the frequency range in the model for determining the Doppler power spectral density. Higher values of Doppler shifts occur when the signal direction is orientated parallelly with respect to the turbines rotors. To put it simply, Doppler shift is maximum when signal reflection is minimum. During the measurements discussed in the paper the opposite scenario was applied, though it was our intention to maximize the reflection by sending the signal perpendicularly to the rotors’ planes. This condition was surely met for the turbines located at the front rows; however, for the turbines in back rows, the signal orientation could gradually “move away” from perpendicular towards parallel. This is the reason why the low power components in Figure 9 exhibit higher Doppler shift compared to the high power components: the former simply represent the turbines further back, while the latter represents those in the front rows. The analysis of the performed tests confirmed the legitimacy of using the model of determining the Doppler power spectral density specified in [11]; however, for frequencies from about −10 Hz to 10 Hz, it is recommended to use the variant for low variability and outside this range to apply the high variability variant. This approach most faithfully reflects the shape of the signal spectrum recorded during measurements for the conditions prevailing at that time and the measured configuration of wind turbines, as shown in Figure 9.

### 3.3. Simulations for Systems Operating in the VHF Band in the Marine Environment of Wind Farms

Based on the ITU-R BT.1893-1 recommendation and the proposed radio channel model, the simulation analyzes of the offshore wind farms’ impact on the VHF systems were carried out. In this section, the assumptions and results of the simulation will be presented. 

In order to investigate the impact of the offshore radio channel on the VHF systems operating near the wind farm, the channel coherence bandwidth and time were first calculated for selected scenarios. The calculations were made for the following parameters: 

Two variant of the turbines’ arrangement—5 × 5 (see Figure 10a) and 10 × 10 (see Figure 10b), differing in the turbine density (the area is the same): 1 km and 2 km distance between the turbines, respectively;

Ship locations (various variants)—ships at 3 distances: 500 m, 2 Nm (2 nautical miles), and 10 Nm from the wind farm;Frequency—161.8 MHz;Average radius of the tower—two variants: 3 m and 4 m;Tower height—150 m;Blade length—115 m;Turbine speed—8 rpm.

Additionally, the height of the ship terminal antenna was assumed to be 10 m above sea level. The mutual location of the transmitter, receiver, and wind farm, as well as the distances between them, were different for each scenario.

For the purposes of the article, the level of the signal received in the analyzed area was simulated, and the CIR parameter value (i.e., Carrier to Interference Ratio) was analyzed for the VHF transmitter located in three distances (Figure 10):

500 m south of the border of the offshore wind farm area (point P1);2 Nm south of the boundary of the offshore wind farm area (point P2);10 Nm south of the boundary of the offshore wind farm area (point P4).

The points P3, P5, and P6 mark the locations of the receiver. These values, based on the recommendations published by the MCA (Maritime and Coastguard Agency) of Great Britain, correspond to three risk levels: high, medium, and low [30]. 

In the context of the following simulations, the term “useful radio range” for interference analyzes means a situation in which the following two criteria are simultaneously met at a given point in the area:

Sensitivity criteria—i.e., the level of the signal received from a given station must be higher than the receiver sensitivity;

Interference criteria—i.e., the actual value of the useful carrier to interference ratio (CIR) must be higher than the assumed value of the minimum required CIR (for the purposes of these simulations, CIR_minimum = 10 dB was assumed).

It should be noted that simulations were carried out for the reference ship’s communication system with the following parameters: 

EIRP power = 41 dBm; Receiver sensitivity: −105 dBm; terminal antenna height: 10 m above sea level. 

These are typical parameters of VHF transmitters used in ships. The results presented in this section were obtained with the use of the radio planning simulation tool described in the article [12]. The presence of the wind farm has been simulated as a scattered source of interference, where each turbine was modelled as a source of reflected (unwanted) power radiating towards individual stations of the communication systems. The theoretical reflected signal level (UFSR) was determined for each turbine using the ITU-R BT.1893-1 Annex 1 or Annex 2 models [11], separately for each scenario. The cumulative impact of the interferences has been simulated using the k-LNM log-normal model [31,32]. This method was developed in order to be used in practice when designing radio networks (especially where the levels of signals measured in the radio channel change over time). It provides very accurate results in a relatively short calculation time, possible to be carried out by network designers. K-LNM is an estimation method used for statistical calculations of the sum of distributions of several log-normally distributed variables (interfering signals from each turbine). To increase the accuracy of the LNM method in regions where high land coverage is required (high probability of coverage), a correction factor *k* is introduced. This method is based on the assumption that the sum of distributions of desirable and undesirable fields is also log-normally distributed. In summary, it is a commonly recognized method often applied for the purpose of designing radio networks and also used in the planning tool implemented by the authors of the article. The ITU-R BT.1893-1 model also describes the standardized equivalent directional characteristic of a wind turbine, which was used in the planning tool to model each turbine (i.e., each interference source) as a hypothetical “antenna” featuring a specific radiation pattern (see Figure 11). In case of using ITU-R BT.1893-1 Annex 1 (reflections from the turbine rotors), the “antenna” representing the interference source was set on turbine mast height, and in case of ITU-R BT.1893-1 Annex 2 (reflection from the turbine mast), the “antenna” representing the interference source was set at a half of the turbine mast height.

The simulations have been conducted using the newest, general-purpose ITU-R P.2001-4 propagation model [33], which by definition is a very universal model for general purpose terrestrial radio systems operating in the frequency range from 30 MHz to 50 GHz. It has a wide range of applicability in frequency, distance, and percentage time with no discontinuities in its output. In particular, it predicts both fading and enhancements of signal level, for signals which may be either wanted or potentially interfering. 

Due to the fact that the channel coherence time depends only on the wind turbine parameters and the operating frequency of the system, its value is the same for all considered scenarios and equals to 4.07 ms. Therefore, the relationship (36) is satisfied, as the symbol duration in the AIS system is 0.104 ms. Thus, in the considered channel, the impulse response does not change during the symbol duration, and only slow fading occurs in the channel. The coherence bandwidth determined during the tests of various simulation scenarios is presented in Table 14. The case 6 is given as an example for a wide coherence bandwidth—such situation occurs when the distance between the transmitter and receiver is less than their distances to the wind turbines.

The coherence bandwidth determined during the tests of various simulation scenarios ranged from 73.748 kHz (for cases of high spatial separation of the transmitter and receiver) to 262.733 kHz (when the transmitter and receiver were close to each other and close to the offshore wind farm). The bandwidth of the AIS system operating in the VHF frequency range is 12.5 kHz and for this system, mainly flat fading will occur. The situation will be different in case of the novel VHF system—VDES [6], which will feature channels with a width up to 100 kHz. For this system, frequency-selective fading will also occur, as the channel coherence bandwidth will in some cases be smaller than the channel bandwidth. However, it should be mentioned here that the VDES system, developed and standardized in the last few years at the IALA and ITU, is a much more advanced system, employing channel equalizers. Moreover, the VDES system allows us to use the narrower transmission bandwidth in difficult conditions (e.g., 25 kHz). When the VDES system is launched in Poland, it will be necessary to take into account the presence of the constructed wind farms.

The simulation results presented in Figure 12 and Figure 13 below show the reference communication system Carrier to Interference Ratio (CIR) plots for different wind farm spacing configurations (25 turbines—5 × 5, and 100 turbines—10 × 10), two turbine tower radius dimensions (3 m and 4 m) and different distance of the ship from the wind farm (500 m, 2 Nm, 10 Nm). The impact of the wind turbine dimension (tower radius) on the ship’s reference communication system is most noticeable for small distances. The higher tower radius dimensions translates into the higher reflected power level according to ITU-R BT.1893-1 Annex 2 model [11], resulting in the higher interferences around the turbines, as can be seen in Figure 12 (*b* vs. *a*, and *d* vs. *c*). The number of reflecting turbines also has a significant impact on the communication system CIR levels—the higher number of turbines the higher cumulative interfering areas, as can be seen in Figure 12 (*c* vs. *a*, and *d* vs. *b*).

The impact of the wind farm on the ship’s reference communication system also depends on the distance from the wind farm, as can be seen in Figure 13. Assuming that the acceptable reference communication system CIR level is 10 dB, then seven of ten areas on the presented CIR plots will represents the areas with no coverage, where the interference criteria is not meet and the continuous radio communication is not possible. The size of other (3 of 10) coverage areas increases as the distance between ship and wind turbines increases, which can be seen in Figure 13 (c vs. b vs. a, and f vs. e vs. d).

It is also worth nothing, that at large distances (10 Nm), both simulated wind farms— the large one (100 turbines 10 × 10) and the smaller one (25 turbines 5 × 5)—have a similar impact on the reference communication system coverage area, as can be seen in Figure 13 (*c* vs. *f*). To sum this up, the most unfavorable results were observed in the following situations:

When the ship was further from the farm;For a farm consisting of a larger number of turbines;For a farm with a larger average tower radius.

While this may seem surprising, it is fully in line with the specifics of the Annex 2 propagation model from the recommendation [11]. According to this document, the size of the effective reflection area of the turbine is directly proportional to the distance from the transmitter (see formula 25). Thus, when the ship was close to the wind farm, the reflecting surface of the turbines (de facto: the mast) was smaller compared to the case when it was located 10 Nm away from the farm. Obviously, the larger the reflecting surface, the higher the potential negative impact of the power plant on the radio signal—which is directly confirmed by the simulation results presented above. The second reason is that the width of the Fresnel zone radius increases with the distance, so in the case of ship-to-ship communication (antennas at a relatively low height compared to the turbine) the attenuation between the transmitter and receiver will always be greater (due to diffraction) than between the receiver and the wind turbine, assuming the same length of the propagation path. Conducting a simulation of radio planning is very important, first of all to check whether the range of the useful signal in the presence of an offshore farm is sufficient and whether necessary corrective actions should be taken to eliminate negative impacts of the wind farm on systems operating in the marine VHF band.

## 4. Conclusions

This article addresses the topic of radio channel modeling for VHF systems in the offshore wind farm propagation environment. Observing the continuous growth of interest in renewable energy sources such as wind, this issue will probably gain more and more importance in the design of onshore and offshore wind farms. The authors, through a measurement campaign in the wind farm environment, confirmed the applicability of the ITU-R BT.1893-1 model (both Annex 1 and Annex 2) in the marine VHF band (160 MHz) at sea. First, as part of the radio channel modeling, we performed the analysis of the signals level reflected from the turbines. In the model included in Annex 1, the basic mechanism of reflection from wind turbines is the reflection from their blades, while in the model from Annex 2 the main mechanism is the reflection from the turbine’s mast. Additionally, the main area of the Annex 2 validity is the scenario when the transmitter and turbine are close to one another. 

In the case of the application of Annex 1, the theoretical level of the UFSR (reflected signal) resulting from the model was on average 15.1 dB higher than it was shown in the measurements. In turn, the model from Annex 2 overstated the level of the reflected signal power by an average of 14.3 dB in relation to the measurement data. As we can see, regardless of the Annex applied, the ITU-R BT.1893-1 recommendation is definitely pessimistic with respect to the VHF maritime band, which means the real level of the unwanted signal reflection from the turbines is less severe than the theoretical values. The pessimistic character of the model is mainly due to following factors:The ITU-R recommendation was developed under the assumption that the propagation paths are model using the simplest possible way, i.e., the free-space loss model, while the authors of this paper employed much more sophisticated model ITU-R P.2001-4 in their simulations. The ITU-R P.2001 is a universal, complicated, general-purpose propagation model valid for radio systems operating in the wide frequency range from 30 MHz to 50 GHz. The approximation of the propagation path conditions obtained using that model are definitely more reliable and closer to reality than those obtained under the free-space loss scenario. The additional diffraction attenuation calculated for the spherical Earth and for the turbines located close to Tx and Rx does not exceed 5 dB, and for the turbines far away from Tx/Rx it can be more than 10 dB. In the future campaigns, we plan to employ high antenna mast at the transmitter and receiver sides (about 10 m), to reduce the impact of diffraction (which is otherwise difficult to estimate during the measurements).The BT.1893-1 recommendation was developed under the assumption that crucial components of the turbines will be modelled as triangle (blade—Annex 1) and cylinder (mast—Annex 2). This is merely an approximation, as the actual components on the existing wind farms are built in a slightly different way (the mast is not an ideal cylinder—it is usually composed of cylindrical sections characterized by varying radii and the blade is not an ideal triangle). Such simplification definitely affects the results provided by the model when compared with real measurement data.During actual measurements, it is extremely difficult to emulate the maximum reflection case (i.e., the worst-case scenario), as it requires a perfectly perpendicular orientation of the transmitter with respect to the rotor’s plane. It is not easy to achieve that, especially due to variable weather conditions (changing direction of wind which moves the rotors, etc.). This issue may clearly affect the results, as even a slight deviation from the 90 degrees requirement (i.e., a few degrees off) noticeably reduces the unwanted reflection level.

All these three factors combined explain why the model may give overly pessimistic data. On the other hand, if we are aware of that and if we determine the relevant “corrective coefficients” (in this case: 15 dB for Annex 1 and 9 dB for Annex 2), than the ITU-R BT.1893-1 model can well be utilized to assess the offshore wind turbines impact on maritime VHF radio systems.

During the study described in the paper, the authors also analyzed the Doppler power spectra density. On the basis of the collected and analyzed measurement data, it should be stated that recommendation ITU-R BT.1893-1 faithfully illustrates the shape of the Doppler power spectra density. For the existing measurement conditions, it is stated that for frequencies from about −10 to +10 Hz the best representation is the low variability version of the Doppler spectrum, while outside this range the best approximation is the high variability version. On the basis of the measurements and their comparison with the theoretical results, knowledge was gained regarding the situations for which it is most advisable to use a given model from the ITU-R BT.1893-1 recommendation. Based on this, it was possible to simulate selected radio channel parameters for ship systems in the marine environment. As a result it was possible to obtain accurate received signal power level and Carrier to Interference Ratio (CIR) values, for different offshore wind farm configurations. These parameters show exactly what coverage can be expected in the selected area and whether there is a need for corrective actions to improve radiocommunication quality in this area. An offshore verification campaign is planned in the future to confirm all assumptions discussed in this article. 

The paper dealt with the offshore wind farms’ impact analysis of the radio systems operating in the maritime VHF band. We should realize that, while VHF is the major maritime frequency range, there are other bands and systems relevant to this topic as well. This includes—among others—the UHF band, radar systems, radionavigation, specific military solutions, etc. In most cases, those area will also have to be addressed while evaluating the impact of the future wind farms. That requires appropriate mathematical and theoretical background, building necessary tools (e.g., for simulations), and gaining full knowledge about the analyses systems. These factors are obviously beyond the scope of this article, but at the same time, they also show how complicated, multi-faceted, and important these issues are for the overall development of the wind energy at sea.

## Figures and Tables

**Figure 1 sensors-23-07593-f001:**
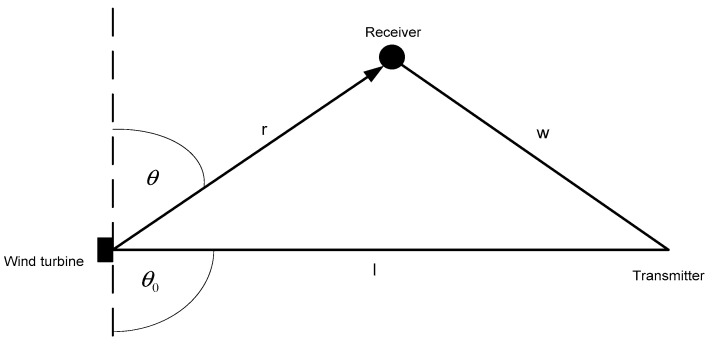
The arrangement of the transmitter, receiver, and wind turbine.

**Figure 2 sensors-23-07593-f002:**
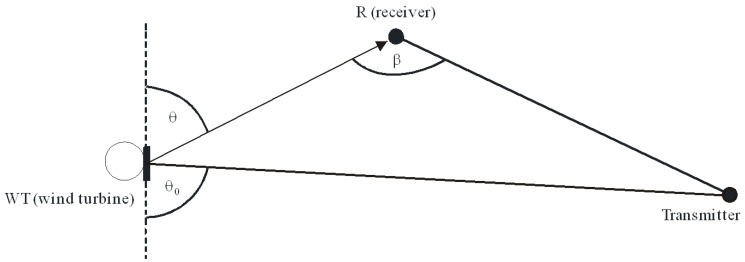
Graphic interpretation of the angles occurring in the dependence (8).

**Figure 3 sensors-23-07593-f003:**
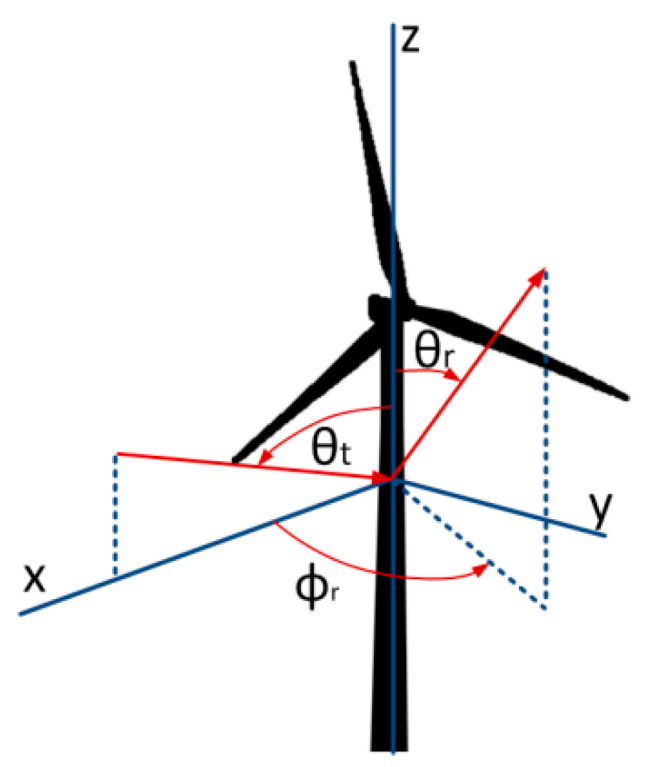
Graphic interpretation of the angles occurring in the dependence (15).

**Figure 4 sensors-23-07593-f004:**
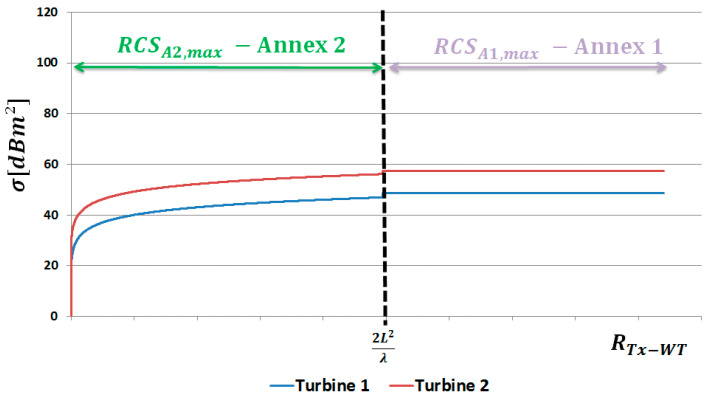
The dependence of the RCS parameter on the distance between the wind turbine and the transmitter.

**Figure 5 sensors-23-07593-f005:**
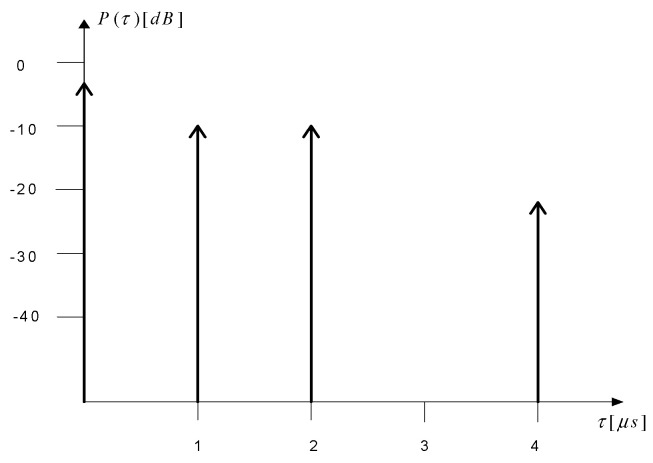
Example of a power delay profile in a radio channel.

**Figure 6 sensors-23-07593-f006:**
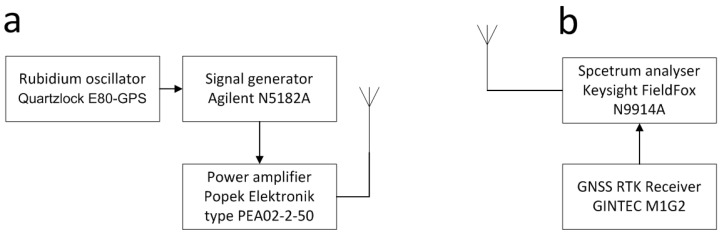
Schematic representation of the measurement set transmitting (**a**) and receiving (**b**).

**Figure 7 sensors-23-07593-f007:**
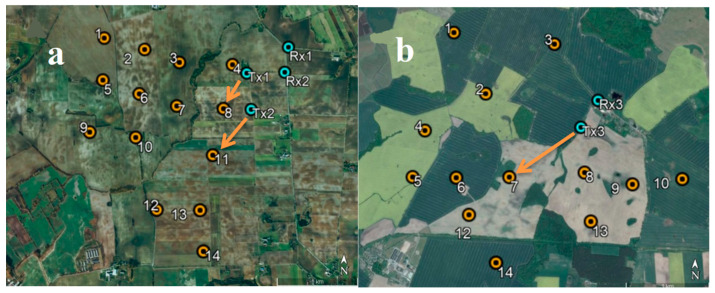
Location of the transmitter (Tx) and receiver (Rx) on a given measurement scenarios (1, 2 and 3) in the back-ground of the north (**a**) and south (**b**) Jasna farms. The orange points indicate wind turbines, while arrows indicate the direction of signal transmission.

**Figure 8 sensors-23-07593-f008:**
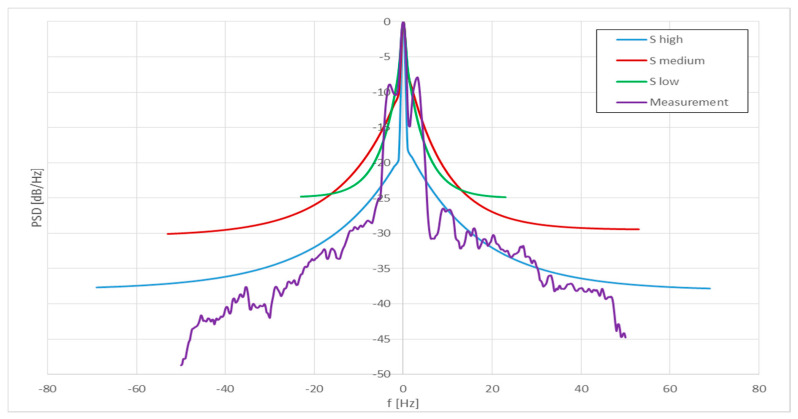
Doppler power spectral density measured on scenario 1 and calculated for the conditions prevailing on that day.

**Figure 9 sensors-23-07593-f009:**
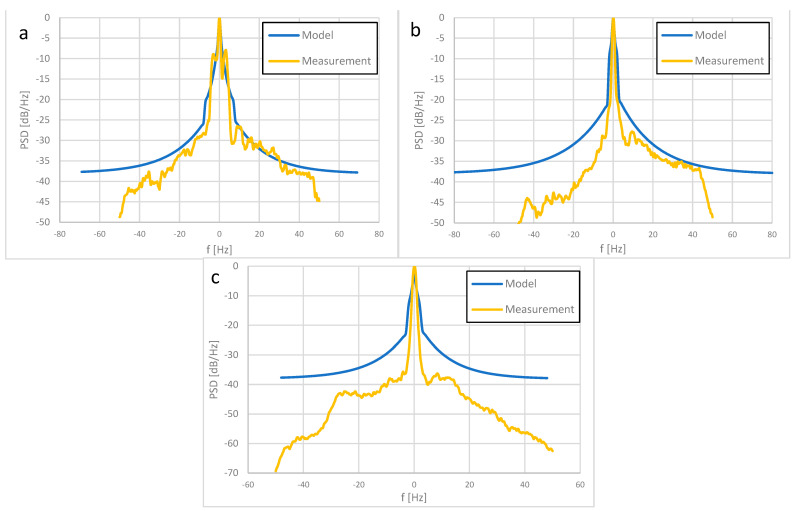
Doppler power spectral density measured on scenario 1 (**a**), scenario 2 (**b**), scenario 3 (**c**) and proposed theoretical model for each case.

**Figure 10 sensors-23-07593-f010:**
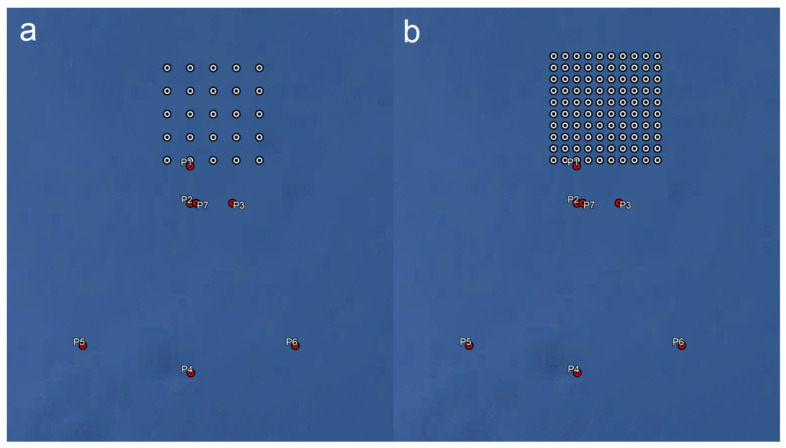
VHF transmitter locations for simulation for 5 × 5 (**a**) and 10 × 10 (**b**) turbine arrangement configurations.

**Figure 11 sensors-23-07593-f011:**
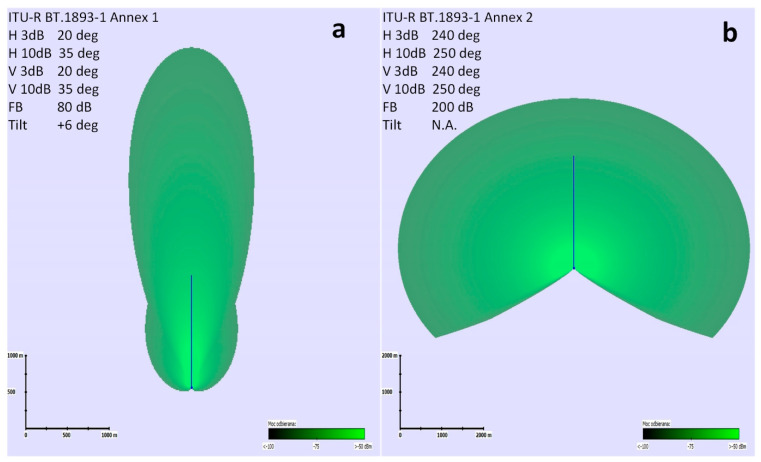
Antenna radiation pattern modeled using ITU-R BT.1893-1 Annex 1 (**a**) and Annex 2 (**b**) for 160 MHz. Remarks: H 3 dB/10 dB—3 dB/10 dB beam width of the antenna in the horizontal plane; V 3 dB/10 dB—3 dB/10 dB beam width of the antenna in the vertical plane; FB—Front-to-back ratio; Tilt—represents the typical wind turbine relation to the vertical axis.

**Figure 12 sensors-23-07593-f012:**
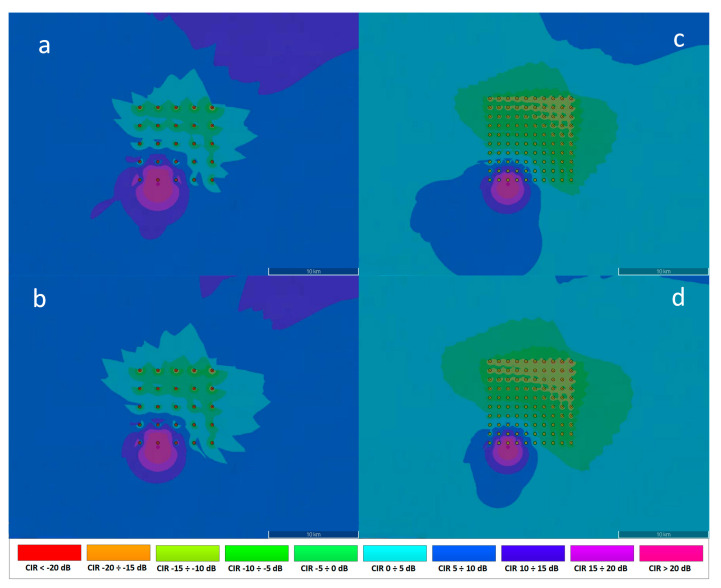
CIR plots for reference communication system transmitting from the ship at a distance of 500 m from the wind farm, for different wind farm configurations and turbine tower radius: (**a**) 25 turbines (5 × 5) and 3 m turbine tower radius; (**b**) 25 turbines (5 × 5) and 4 m turbine tower radius; (**c**) 100 turbines (10 × 10) and 3 m turbine tower radius; (**d**) 100 turbines (10 × 10) and 4 m turbine tower radius.

**Figure 13 sensors-23-07593-f013:**
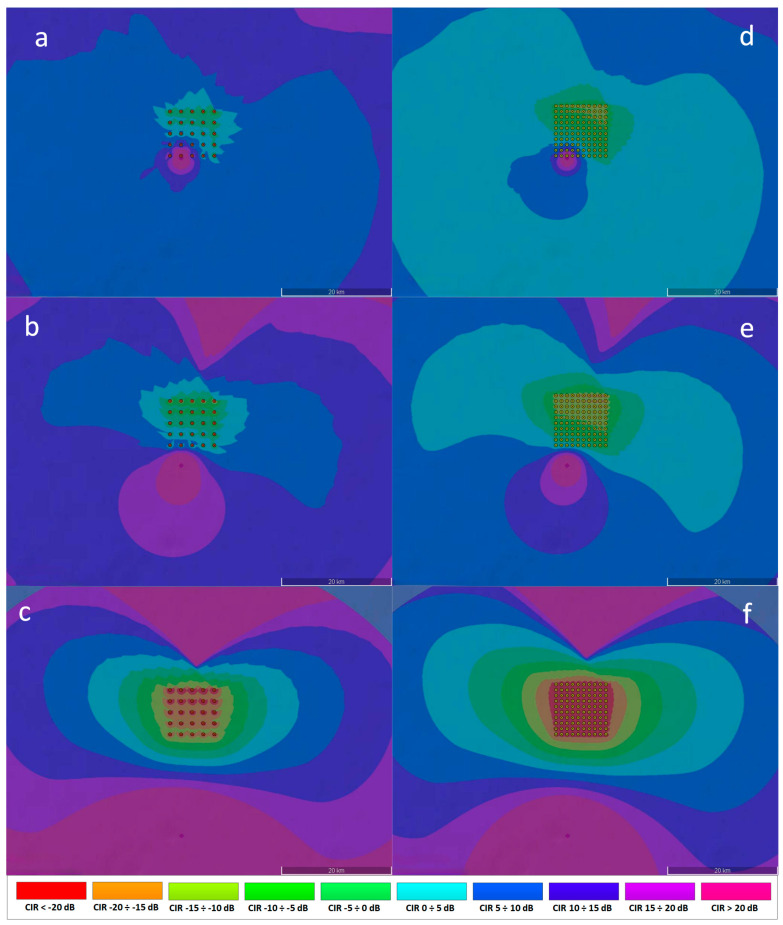
CIR plots for reference communication system transmitting from the ship at different distances from the wind farm, for different wind farm configurations and 3 m turbine tower radius: (**a**) 25 turbines (5 × 5) and ship at distance of 500 m; (**b**) 25 turbines (5 × 5) and ship at distance of 2 Nm; (**c**) 25 turbines (5 × 5) and ship at distance of 10 Nm; (**d**) 100 turbines (10 × 10) and ship at distance of 500 m; (**e**) 100 turbines (10 × 10) and ship at distance of 2 Nm; (**f**) 100 turbines (10 × 10) and ship at distance of 10 Nm.

**Table 1 sensors-23-07593-t001:** Parameters of the Jasna wind turbines for which measurements were made (based on the data from [25,26,27] and the farm operator’s data).

Wind farm	Jasna	
Part of the farm	North	South
Number and types/models of turbines on the farm	25 Vestas turbines V126, 3.3 MW/3.45 MW	14 Vestas turbines V126, 3.3 MW/3.45 MW
Turbine’s tower height [m]	117	137
Material (tower)	Steel	
Length of the blades [m]	61.7	
Max. Blade width [m]	4	
Area of a single blade [m^2^]	123.4	
Number of the blades	3	
Total area of the blades [m^2^]	370.2	
Material (blades)	GFRP (Glass Fiber Reinforced Polymer)	
Average radius of the tower	3.4	
Turbines’ layout (arrangement)	Non-uniform	
Speed range [rpm]	5.9–16.3	
Distance between the turbines [m]	450–550	

**Table 2 sensors-23-07593-t002:** Distances between transmitter–receiver–wind turbine and theoretical signal strength parameters for all measurement scenarios (see Figure 7).

	Tx-Rx [m]	Wind Turbine-Tx [m]	Wind Turbine-Rx [m]	FSR (Theoretical Value) [dBm]	Max. Theoretical UFSR (Annex 1) [dBm]	Max. Theoretical UFSR (Annex 2) [dBm]
Scenario 1	650	550	1150	−19.5	−50.38	−51.15
Scenario 2	650	750	1400	−19.5	−54.78	−55.55
Scenario 3	400	1050	1450	−15.3	−58.01	−58.78

**Table 3 sensors-23-07593-t003:** Pulse signal in the time domain in the 161 MHz band (north Jasna farm, 26 October 2021 (Scenario 1); north Jasna farm, 27 October 2021 (Scenario 2); and south Jasna, 26 October 2021 (Scenario 3)).

	Measurement	Model
Scenario 1	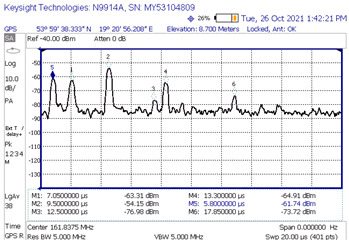	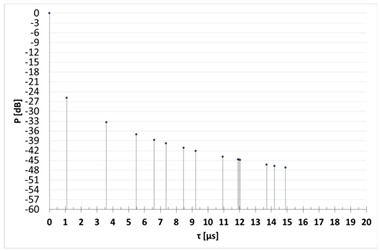
Scenario 2	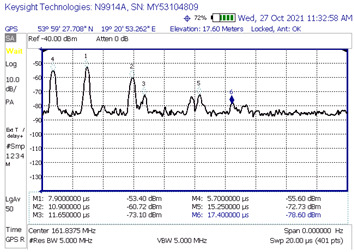	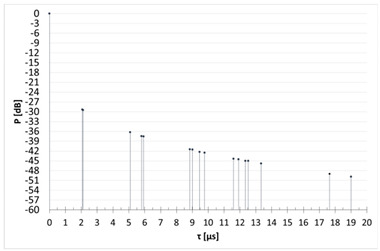
Scenario 3	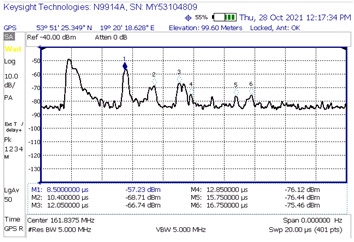	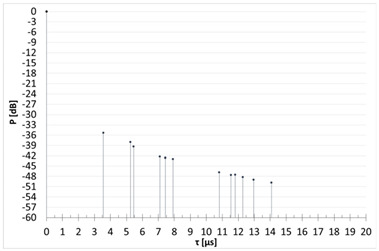

**Table 4 sensors-23-07593-t004:** Comparison of the measured and theoretical values of the power of the direct signal (FSR) and the reflected signal (UFSR)—scenario 1.

The Peak in Table 3	Turbine Number from Which the Signal Was Reflected	Measurement Considering the Transmitting Antenna Characteristics [dBm]	Theory—Annex 1 [dBm]	Theory—Annex 2 [dBm]	Measurement—Theory Annex 1 Difference [dB]	Measurement—Theory Annex 2 Difference [dB]	Comments
Direct Component (FSR)							
5	N/A (direct signal)	−61.74	−19.5	−19.5	−42.24	−42.24	The theoretical value of the direct component FSR does not depend on the adopted model (Annex1/Annex2)
Reflected Components (UFSR)							
1	4	−57.13	−69.53	−44.14	+12.40	−12.99	
2	8	−54.15	−43.51	−51.86	−10.64	−2.29	
3	7	−76.21	−98.54	−57.55	+22.33	−18.66	
4	11	−64.83	−51.49	−58.24	−13.34	−6.59	
6	13	−73.57	−61.13	−63.15	−12.44	−10.42	*
6	1	−69.79	−123.16	−63.18	+53.37	−6.61	
6	5	−71.46	−91.03	−63.28	+19.57	−8.18	
7	2	−74.13	−107.71	−59.6	+33.58	−14.53	

NOTE: the sign (—) in the columns “Measurement-theory difference…” means that the model shows a higher signal level than the measured value (i.e., the model is pessimistic); the sign (+) means the opposite situation (optimistic model). * The signals reflected from the turbines 1, 5, and 13 are only 120 ns apart, i.e., less than half the pulse duration (250 ns). Thus, peak 6 probably represents a situation where the signals reflected from these three turbines overlapped.

**Table 5 sensors-23-07593-t005:** Comparison of the measured and theoretical values of the power of the direct signal (FSR) and the reflected signal (UFSR)—scenario 2.

The Peak in Table 3	Turbine Number from Which the Signal Was Reflected	Measurement Considering the Transmitting Antenna Characteristics [dBm]	Theory—Annex 1 [dBm]	Theory—Annex 2 [dBm]	Measurement—Theory Annex 1 Difference [dB]	Measurement—theory Annex 2 Difference [dB]	Comments
Direct Component (FSR)							
4	N/A (direct signal)	−55.6	−19.5	−19.5	−36.1	−36.1	The theoretical value of the direct component FSR does not depend on the adopted model (Annex1/Annex2)
Reflected Components (UFSR)							
1	8	−51.34	−70.47	−48.11	+19.13	−3.23	*
1	4	−42.20	−88.41	−48.14	+46.21	+5.94	
2	11	−60.72	−45.81	−54.87	−14.91	−5.85	
3	7	−70.94	−75.67	−56.03	+4.73	−14.91	**
3	3	−67.57	−116.37	−56.08	+48.80	−11.49	
7	2	−71.19	−137.35	−60.1	+66.16	−11.09	
5	13	−72.60	−59.01	−60.83	−13.59	−11.77	
6	12	−78.59	−57.73	−62.96	−20.86	−15.63	

NOTE: the sign (—) in the columns “Measurement-theory difference…” means that the model shows a higher signal level than the measured value (i.e., the model is pessimistic); the sign (+) means the opposite situation (optimistic model). * The signals reflected from the turbines 4 and 8 are only 62 ns apart, which is four times less than the pulse duration (250 ns). Therefore, peak No. 1 probably represents a situation in which the signals reflected from these two turbines overlapped. ** The signals reflected from the turbines 3 and 7 are only 140 ns apart, so less than the pulse duration (250 ns). Therefore, peak No. 3 probably represents a situation where the signals reflected from these two turbines overlapped.

**Table 6 sensors-23-07593-t006:** Comparison of the measured and theoretical values of the power of the direct signal (FSR) and the reflected signal (UFSR)—scenario 3.

The Peak in Table 3	Turbine Number from Which the Signal Was Reflected	Measurement Considering the Transmitting Antenna Characteristics [dBm]	Theory—Annex 1 [dBm]	Theory—Annex 2 [dBm]	Measurement—Theory Annex 1 Difference [dB]	Measurement—Theory Annex 2 Difference [dB]	Comments
Direct Component (FSR)							
7	N/A (direct signal)	−49	−15.3	−15.3	−33.7	−33.7	The theoretical value of the direct component FSR does not depend on the adopted model (Annex1/Annex2)
Reflected Components (UFSR)							
1	8	−54.48	−73.35	−48.73	+18.87	−5.75	
2	3	−59.34	−126.63	−51.32	+67.29	−8.02	*
2	9	−61.24	−108.12	−52.59	+46.88	−8.65	
3	7	−66.74	−49.27	−55.52	−17.47	−11.22	
4	10	−64.99	−157.23	−56.28	+92.24	−8.71	
5	6	−76.31	−59.03	−60.11	−17.28	−16.20	
6	1	−71.31	−132.26	−60.81	+60.95	−10.50	

NOTE: the sign (—) in the columns “Measurement-theory difference…” means that the model shows a higher signal level than the measured value (i.e., the model is pessimistic); the sign (+) means the opposite situation (optimistic model). * The signals reflected from the turbines 3 and 9 are only 200 ns apart, hence less than the pulse duration (250 ns). Peak 2, therefore, represents the probable situation where the signals reflected from these two turbines overlapped.

**Table 7 sensors-23-07593-t007:** Comparison of theoretical and measured coherence bandwidths for all measurement scenarios.

	Theoretical Coherence Bandwidth [kHz]	Measured Coherence Bandwidth [kHz]	Measured Coherence Bandwidth after Substituting the Theoretical FSR Value [kHz]
Scenario 1	632.452	105.356	2390.196
Scenario 2	608.537	108.365	2760.748
Scenario 3	1017.126	122.867	5083.339

**Table 8 sensors-23-07593-t008:** Rotational speeds of rotors and Doppler frequencies on individual measurement scenarios.

	Location	Rotational Speed of the Rotor of the Nearest Turbine during the Measurements	Maximum Doppler Frequency [Hz]	Channel Coherence Time [ms]
Scenario 1	53°59′38.329″ N 19°20′56.212″ E	10.9 rpm	75.99	5.567
Scenario 2	53°59′27.717” N 19°20′53.273″ E	12.7 rpm	88.53	4.778
Scenario 3	53°51′25.342″ N 19°20′18.622″ E	7.6 rpm	52.98	7.984

**Table 9 sensors-23-07593-t009:** Limit frequencies from ITU-R BT.1893-1 model of the Doppler power spectral density for the measurement scenarios.

Limit Frequency from ITU-R BT.1893-1 Model	Scenario 1	Scenario 2	Scenario 3
0.9 $f_{B\;max}$ (High variability)	68.39	79.68	47.68
0.7 $f_{B\;max}$ (Medium variability)	53.19	61.97	37.09
0.3 $f_{B\;max}$ (Low variability)	22.8	26.56	15.89

**Table 10 sensors-23-07593-t010:** Doppler power spectral density [dB/Hz]—scenario 1.

f [Hz]	−30	−20	−10	−5	−3	−1	1	3	5	10	20	30
Low	−	−24.7	−22.7	−17.8	−13.6	−6.8	−6.6	−13.4	−17.7	−22.7	−24.8	−
Medium	−28.4	−26	−20.5	−15.7	−13.1	−10.1	−7.1	−11.7	−15.3	−21.5	−27	−28.7
High	−34.7	−32	−27.1	−23.3	−21.5	−19.4	−18	−20.4	−22.5	−26.7	−32.2	−34.9
Measurement	−42	−33.8	−29.2	−24.9	−8.9	−8.5	−12.1	−8	−20.9	−26.7	−30.6	−34.7

**Table 11 sensors-23-07593-t011:** Doppler power spectral density [dB/Hz]—scenario 2.

f [Hz]	−30	−20	−10	−5	−3	−1	1	3	5	10	20	30
Low	−	−24.5	−21.9	−16.6	−12.4	−6.2	−6	−12.2	−16.4	−21.8	−24.6	−
Medium	−27.6	−24.9	−19.4	−14.8	−12.5	−9.9	−6.7	−10.8	−14.1	−20.1	−26	−28.3
High	−33.7	−30.9	−26.1	−22.7	−21.1	−19.3	−17.8	−19.9	−21.8	−25.7	−30.9	−33.9
Measurement	−42.6	−41.2	−29.2	−24.9	−8.9	−8.5	−12.1	−8	−20.9	−28	−33.8	−36.1

**Table 12 sensors-23-07593-t012:** Doppler power spectral density [dB/Hz]—scenario 3.

f [Hz]	−30	−20	−10	−5	−3	−1	1	3	5	10	20	30
Low	−	−	−24.1	−20.7	−16.6	−8.6	−8.4	−16.4	−20.6	−24.2	−	−
Medium	−29.7	−28.2	−23.4	−18	−14.8	−10.8	−8.2	−14.2	−18.5	−24.6	−28.6	−29.3
High	−36.5	−34.4	−29.6	−25.1	−22.7	−19.9	−18.6	−21.8	−24.5	−29.5	−34.6	−36.7
Measurement	−48.3	−43.6	−40.7	−38.4	−36.2	−14.6	−10.3	−35.5	−40	−38	−44.8	−51.1

**Table 13 sensors-23-07593-t013:** The mean difference in UFSR determination with the use of both analyzed models.

Model	Mean Difference (Error) [dB]
ITU-R BT.1893-1 Annex 1	−15.07
ITU-R BT.1893-1 Annex 2	−9.16

**Table 14 sensors-23-07593-t014:** Coherence bandwidth depending on the radius of the tower and the configuration of wind turbine spacing.

ID	Distance of the Transmitter from the Wind Farm (see Figure 10)	Distance of the Receiver from the Wind Farm (see Figure 10)	Distance between Transmitter and Receiver	Coherence Bandwidth for the Selected Tower Radius for a 5 × 5 Turbine Configuration [kHz]		Coherence Bandwidth for the Selected Tower Radius for a 10 × 10 Turbine Configuration [kHz]	
				Tower Radius		Tower Radius	
				3 m	4 m	3 m	4 m
1	500 m (P1)	2 Nm (P3)	1,7 Nm	262.733	227.648	178.468	155.227
2	500 m (P1)	10 Nm (P5)	10 Nm	73.748	63.947	134.112	116.221
3	2 Nm (P2)	2 Nm (P3)	2 Nm	224.176	194.19	211.424	183.201
4	2 Nm (P2)	10 Nm (P6)	8 Nm	123.083	106.633	134.112	116.221
5	10 Nm (P4)	10 Nm (P6)	10 Nm	102.816	89.057	102.751	89.009
6	2 Nm (P2)	2 Nm (P7)	500 m	1282.658	1110.82	1084.674	939.366

## Data Availability

All collected measurement data and simulation results analyzed during the current study are available from the corresponding author on reasonable request.

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
