# Peer review of "Radio Channel Modelling for VHF System Operating in the Offshore Wind Farms Propagation Environment"

_sensors, 2023, doi:10.3390/s23177593_

Round 1

Reviewer 1 Report

Please see the attached comments.

Minor editing of English language is needed.

Reviewer 2 Report

  • The author needs to include the results of their study rather than referring to the location where the results are available in the manuscript.
  •  In the introduction section of the manuscript, the author should identify the study's contribution through bullet marking.
  • The authors need to cite the source of all the equations they have not proposed.
  • Please use conventional notation to represent decimal points (Equation 13)
  • Can the authors include something similar to Figure 8 for the other scenarios?
  • Please use Figure instead of Fig. (following the journal's style).
  • After the equation, defining the notation starts like a new paragraph by allowing space before the first sentence. In most cases, a new paragraph may not form there; the authors must address it.

Reviewer 3 Report

The article presents the results of research in the field of experimental study of the propagation conditions of radio waves in real communication channels, traditional for wireless communication. The results of measurements of the propagation of VHF radio waves in a wind farm are presented. This is an important research area for practical wireless applications. The authors presented a large amount of experimental material. The material is presented well and clearly. The results of measurements and their analysis will be of interest to specialists in the field of wireless communication. Such practical results are always interesting.

The only remark concerns the design of some tables with measurement results. It seems to me that they can be made more compact (some of them are very sparse and are located on two pages, not one) and more convenient for readers.

The content of the article can be accepted in its current form.
